# Mitochondrial RNA cytosolic leakage drives the SASP

Stella Victorelli [1,2,10] ✉, Madeline Eppard[1,2,10], Hélène Martini [1,2], Seung-Hwa Woo [1,2], Stacia P. A. Everts [1,2], Gung Lee [1,2], Nicholas Pirius[1,2], Nuan Han [1,2,3], Eugene Y. Liang[1,2,4], Ana Catarina Franco[1,2], Yeaeun Han [1,2], Dominik Saul [2,5], Eva Nóvoa[6,7], Rubén Nogueiras [6,7,8], Patrick L. Splinter[9], Steven P. O'Hara[9], Olivia Morgenthaler [9], Lucía Valenzuela-Pérez [9], Hyun Se Kim Lee [9], Diana Jurk [1,2], Nicholas F. LaRusso [9], Petra Hirsova [9] & João F. Passos [1,2] ✉

Senescent cells secrete proinflammatory factors known as the senescence-associated secretory phenotype (SASP), contributing to tissue dysfunction and aging. Mitochondrial dysfunction is a key feature of senescence, influencing SASP via mitochondrial DNA (mtDNA) release and cGAS/STING pathway activation. Here, we demonstrate that mitochondrial RNA (mtRNA) also accumulates in the cytosol of senescent cells, activating RNA sensors RIG-I and MDA5, leading to MAVS aggregation and SASP induction. Inhibition of these RNA sensors significantly reduces SASP factors. Furthermore, BAX and BAK play a key role in mtRNA leakage during senescence, and their deletion diminishes SASP expression in vitro and in a mouse model of Metabolic Dysfunction-Associated Steatohepatitis (MASH). These findings highlight mtRNA's role in SASP regulation and its potential as a therapeutic target for mitigating age-related inflammation.

Cellular senescence is an irreversible growth arrest triggered by stressors, involving activation of p53/p21[CIP1] and p16[INK4A/RB] pathways[1–3]. Senescent cells secrete proinflammatory cytokines, chemokines, and extra-cellular matrix degrading proteins, which are collectively known as the senescence-associated secretory phenotype (SASP)[4,5]. Chronic exposure to the SASP can lead to tissue dysfunction[6]. Furthermore, senescence has been shown to occur in multiple tissues during aging and in multiple chronic diseases[7–10]. Notably, clearance of senescent cells has been shown to improve several age-related conditions[11–13].

Mitochondrial dysfunction is a key feature of cellular senescence. To functionally investigate the role of mitochondria during senescence, we utilized the property of experimentally induced PINK1/Parkin to induce widespread mitophagy to generate human fibroblasts without mitochondria[14]. Senescent cells with no mitochondria had no SASP but still underwent the irreversible cell cycle arrest[15,16]. This led us to propose that mitochondria play a key role in the regulation of the SASP and may be promising targets for anti-senescence therapies that suppress the detrimental SASP, while maintaining the tumor suppressor capabilities of senescent cells[15,16]. We recently found that mitochondria influence the SASP by releasing mitochondrial DNA (mtDNA) into the cytosol, activating the cGAS/STING pathway[17].

[1]Department of Physiology and Biomedical Engineering, Mayo Clinic, Rochester, MN, USA. [2]Robert and Arlene Kogod Center on Aging, Mayo Clinic, Rochester, MN, USA. [3]Molecular Pharmacology and Experimental Therapeutics Graduate Program, Mayo Clinic Graduate School of Biomedical Sciences, Mayo Clinic, Rochester, MN, USA. [4]Mayo Clinic Medical Scientist Training Program, Mayo Clinic, Rochester, MN, USA. [5]Robert Bosch Center for Tumor Diseases, Stuttgart, Germany. [6]Department of Physiology, CIMUS, University of Santiago de Compostela, Santiago de Compostela, Spain. [7]CIBER Fisiopatología de la Obesidad y Nutrición (CIBERobn), Santiago de Compostela, Spain. [8]Galicia Agency of Innovation (GAIN), Xunta de Galicia, Santiago de Compostela, Spain. [9]Division of Gastroenterology and Hepatology, Mayo Clinic, Rochester, MN, USA. [10]These authors contributed equally: Stella Victorelli, Madeline Eppard. ✉e-mail: Victorelli.Stella@mayo.edu; passos.joao@mayo.edu

Mitochondrial RNA (mtRNA), if present in the cytosol, is highly immunogenic likely due to the absence of nucleoside modifications[18]. Abnormal accumulation of endogenous mitochondrial double-stranded RNA (mtdsRNA) in the cytosol has been shown to trigger antiviral signaling and induce a type I interferon response, leading to inflammation[19]. Recent studies have demonstrated that RNA viruses, such as SARS-CoV-2, can trigger senescence and induce the SASP[20,21].

In this study, we sought to investigate the role of mtdsRNA in the regulation of the SASP during cellular senescence. We found that senescent cells accumulate cytosolic mtdsRNA, which activates RNA sensors RIG-I and MDA5, leading to Mitochondrial Antiviral Signaling protein (MAVS) aggregation. Inhibiting RIG-I, MDA5, and MAVS suppresses the SASP, highlighting the role of mtdsRNA in SASP regulation and its potential as a therapeutic target. We further demonstrate that mtdsRNA release depends on the pro-apoptotic proteins BAX and BAK. In vivo, genetic deletion of BAX and BAK, or MAVS, in a mouse model of metabolic dysfunction–associated steatohepatitis (MASH) led to a reduction of the SASP and improved functional parameters in livers. Together, these findings uncover a mitochondria-to-cytosol RNA signaling axis that promotes senescence-associated inflammation and may offer new therapeutic opportunities.

## Results

### Senescent cells have increased cytosolic mtdsRNA and RNA sensor signaling

Using dual immunostaining of the mitochondria outer membrane protein TOM20 and J2 antibody, which recognizes double stranded RNA (dsRNA), visualized by super-resolution Airyscan confocal microscopy, we observed an increased frequency of dsRNA nucleoids in the cytosol of senescent fibroblasts (Fig. 1a, b). Subsequent subcellular fractionation of senescent cells (Fig. 1c) revealed elevated levels of mitochondrial RNA in the cytosolic fraction. Specifically, there was an increase in cytosolic mitochondrial-derived RNA transcripts, including MT-ND5, MT-ND6, and MT-CYB and MT-COI (Fig. 1d). This rise in cytosolic mitochondrial-derived RNA was accompanied by a significant upregulation of cytosolic RNA sensors RIG-I, MDA5, and TLR3 at both the mRNA and protein levels in senescent compared to proliferating cells (Fig. 1e–g). These findings were consistent irrespective of the senescence-inducing stimulus, including in replicative senescent cells (Fig. 1h, i· and Supplementary Fig. 1a–c) and cells induced to senescence by the chemotherapy drugs doxorubicin and etoposide (Supplementary Fig. 1d–h), and in a different cell line (IMR90 fibroblasts) (Supplementary Fig. 1i–l).

Furthermore, we observed that expression of RNA sensors Rig-I, Mda5, and Tlr3 significantly increased with age in murine kidney, heart, liver, and spleen, concomitantly with increased expression of senescence-associated markers p16 and p21. Notably, across these tissues, RNA sensor expression exhibited a positive correlation with the expression of p21, p16, and several SASP factors (Supplementary Fig. 2a–t). These findings suggest a mechanistic link between cellular senescence, the upregulation of RNA sensors, and the SASP.

### Cytosolic mtdsRNA is a driver of the SASP

Having observed an increase in cytosolic mtdsRNA in senescent cells, we investigated whether mtdsRNA alone is sufficient to drive the SASP. To this end, we transfected proliferating human fibroblasts with mtRNA and found a significant rise in expression of common SASP factors, along with elevated levels of RNA sensors MDA5, RIG-I, and TLR3 (Fig. 2a–c).

To further explore the role of mtRNA in inducing the SASP in senescent cells, we created cells without mitochondria to study the role of mtRNA in SASP regulation independently of other mitochondrial functions[14]. We stably transduced human fibroblasts with YFP-Parkin and induced senescence via X-ray irradiation (Fig. 2d). Parkin-expressing senescent cells were treated with the mitochondrial

uncoupler CCCP, triggering widespread mitophagy and generating mitochondria-free cells[14]. Western blot analysis confirmed the absence of mitochondrial proteins NDUFB8, UQCRC2, and COXIV following Parkin-mediated mitophagy (Fig. 2e). qPCR analysis showed that cytosolic mtRNA transcripts were abrogated in these mitochondria-devoid senescent cells (Fig. 2f), with a significant reduction in expression of RNA sensors MDA5 and RIG-I (Fig. 2g).

To assess the functional impact of mtRNA, we transfected mitochondria-free senescent cells with purified mtRNA and performed RNA sequencing (Fig. 2h). Gene set enrichment analysis revealed that IFNB1 target genes (Hecker IFNB1 targets) and NF-κB (TNFA signaling via NF-κB) were active in senescent cells, suppressed following mitochondrial clearance, and partially restored by mtRNA transfection (Fig. 2i, j and Supplementary Fig. 3a, b). Similar trends were observed using the SenMayo dataset (Supplementary Fig. 3c,d). qPCR confirmed that mtRNA transfection partially restored the expression of SASP factors IL1- α, IL6, and IL8 in mitochondria-free senescent cells (Supplementary Fig. 3e).

Because mtDNA can also partially restore inflammatory gene expression in senescent cells lacking mitochondria[17], we directly compared the effects of introducing mtRNA or mtDNA into these mitochondria-depleted cells (Supplementary Fig. 4a). Across a broad panel of inflammation-related genes, including IFNB1 and NF-κB targets, some SASP genes were activated by both nucleic acids, while others responded only to mtRNA or only to mtDNA. This indicates that cytosolic mtRNA and mtDNA each trigger distinct parts of the inflammatory program (Supplementary Fig. 4b, c). A previously defined mtDNA stress signature[22] was upregulated in senescent cells but reversed by mitochondrial clearance. Reintroduction of mtDNA restored and even amplified this signature, whereas mtRNA had minimal effect (Supplementary Fig. 4d).

Next, we sought to prevent mtRNA leakage during senescence by treating cells with an inhibitor of mitochondrial RNA polymerase (POLRMT), the enzyme responsible for transcribing mtDNA into RNA within mitochondria. Treating senescent cells with POLRMT inhibitor IMT1 significantly reduced cytosolic mtdsRNA levels but did not reduce levels of cytosolic DNA (Supplementary Fig. 5a, b). In addition, it reduced the expression of RNA sensors RIG-I, MDA5, TLR3 (Supplementary Fig. 5c), and several SASP components (Supplementary Fig. 5d–g), without impacting on p21 and p16 expression (Supplementary Fig. 5h). Collectively, these findings support mtdsRNA as a driver of the SASP.

### RNA sensing signaling contributes to the development of the SASP

Having established that cytosolic mtRNA can contribute to the SASP, we sought to investigate the underlying mechanisms. Cytosolic RNA sensors are crucial components of the innate immune system, responsible for detecting dsRNA, which is often a sign of viral infection. These sensors ultimately converge into pro-inflammatory signaling pathways, such as NF-κB, interferon-regulatory factors (IRFs), and the NLRP3 inflammasome, leading to the induction of type-I interferon genes and pro-inflammatory cytokines[23–25].

To investigate the role of RNA sensors in cellular senescence and their impact on the SASP, we first treated senescent cells with a TLR3/dsRNA complex inhibitor. However, this treatment did not affect the expression of common SASP components or cyclin-dependent kinase inhibitors p15, p16 and p21 (Supplementary Fig. 6a–d). These results were confirmed using siRNA-mediated knockdown of TLR3 (Supplementary Fig. 6e–g). Based on these results, we concentrated our subsequent efforts on RIG-I and MDA5 as the primary RNA sensors mediating the effects of mtdsRNA on the SASP.

To directly assess whether mtRNA engages these sensors during senescence, we immunoprecipitated RIG-I and MDA5 from both

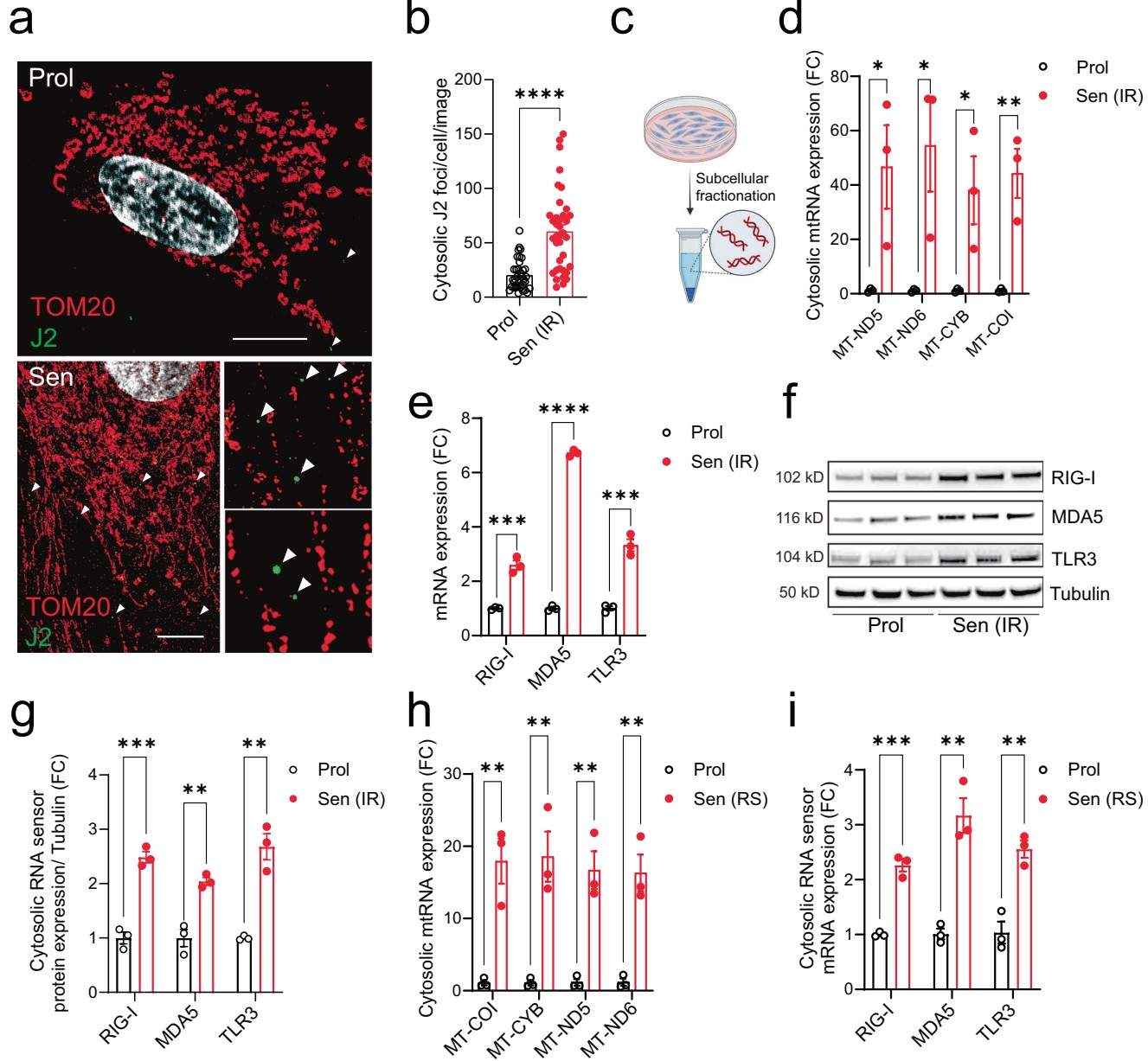

**Fig. 1 | Cytosolic mtRNA leakage is a feature of senescent cells. a** Representative super-resolution Airyscan images of MRC5 human fibroblasts stained for TOM20 (mitochondria, red) and J2 (double-stranded RNA, green) in proliferating and senescent (irradiated; IR) cells. dsRNA foci are enriched in the cytoplasm of senescent cells. Scale bar=10 μm. **b** Quantification of cytosolic dsRNA foci *per* cell in proliferating and senescent (IR) MRC5 fibroblasts. $n = 37$ cells examined over 4 independent experiments. p < 0.0001. **c** Schematic of the subcellular fractionation method used to isolate cytosolic RNA. Created in BioRender. Victorelli, S. (2025) https://BioRender.com/n8s2am6. qPCR quantification of (**d**) cytosolic mitochondrial RNA (mtRNA) transcripts and (**e**) cytosolic RNA sensor transcripts (RIG-I, MDA5, and TLR3) in proliferating and senescent (IR) MRC5 fibroblasts. $n = 3$

independent experiments. **f** Representative Western blot showing protein levels of RNA sensors (RIG-I, MDA5 and TLR3) in proliferating and senescent (IR) MRC5 fibroblasts. **g** Quantification of the Western blots shown in (**f**). $n = 3$ independent experiments. qPCR analysis of (**h**) cytosolic mtRNA transcripts and (**i**) cytosolic RNA sensors in proliferating and replicatively senescent (RS) MRC5 human fibroblasts. $n = 3$ independent experiments. Data are mean ± s.e.m. Statistical significance was assessed by a two-sided Student's unpaired *t*-test (**b**, **d**, **e**, **g**–**i**). **d** $p = 0.0413$, $p = 0.0347$, $p = 0.0421$, $p = 0.0088$; **e** $p = 0.0008$, $p < 0.0001$, $p = 0.0006$; **g** $p = 0.0006$, $p = 0.0039$, $p = 0.0022$; **h** $p = 0.0058$, $p = 0.0075$, $p = 0.0042$, $p = 0.0043$; **i** $p = 0.0004$, $p = 0.0029$, $p = 0.004$.

proliferating and senescent cells and performed qPCR to detect associated mitochondrial RNA transcripts. We found that both RIG-I and MDA5 exhibited significantly increased binding to mtRNA in senescent cells compared to proliferating cells (Fig. 3a–c), supporting a direct role for these sensors in sensing cytosolic mtRNA during senescence.

Having observed the binding of RIG-I and MDA5 to mtRNA in senescent cells, we utilized CRISPR/Cas9 gene editing to generate human fibroblasts deficient in RIG-I and MDA5 (Fig. 3d, e). RNA-

sequencing revealed that the deletion of RIG-I or MDA5 significantly reduced the expression of several SASP components (Fig. 3f, g). Additionally, deletion of RIG-I and MDA5 resulted in a negative enrichment score for established senescence-associated gene sets, SenMayo[26] and SenSig[27], which encompass many SASP factors (Supplementary Fig. 7a–f). It also significantly downregulated Gene Ontology (GO) terms related to inflammatory processes (Supplementary Fig. 7g, h) and reduced expression of several interferon-related genes (Supplementary Fig. 7i, j).

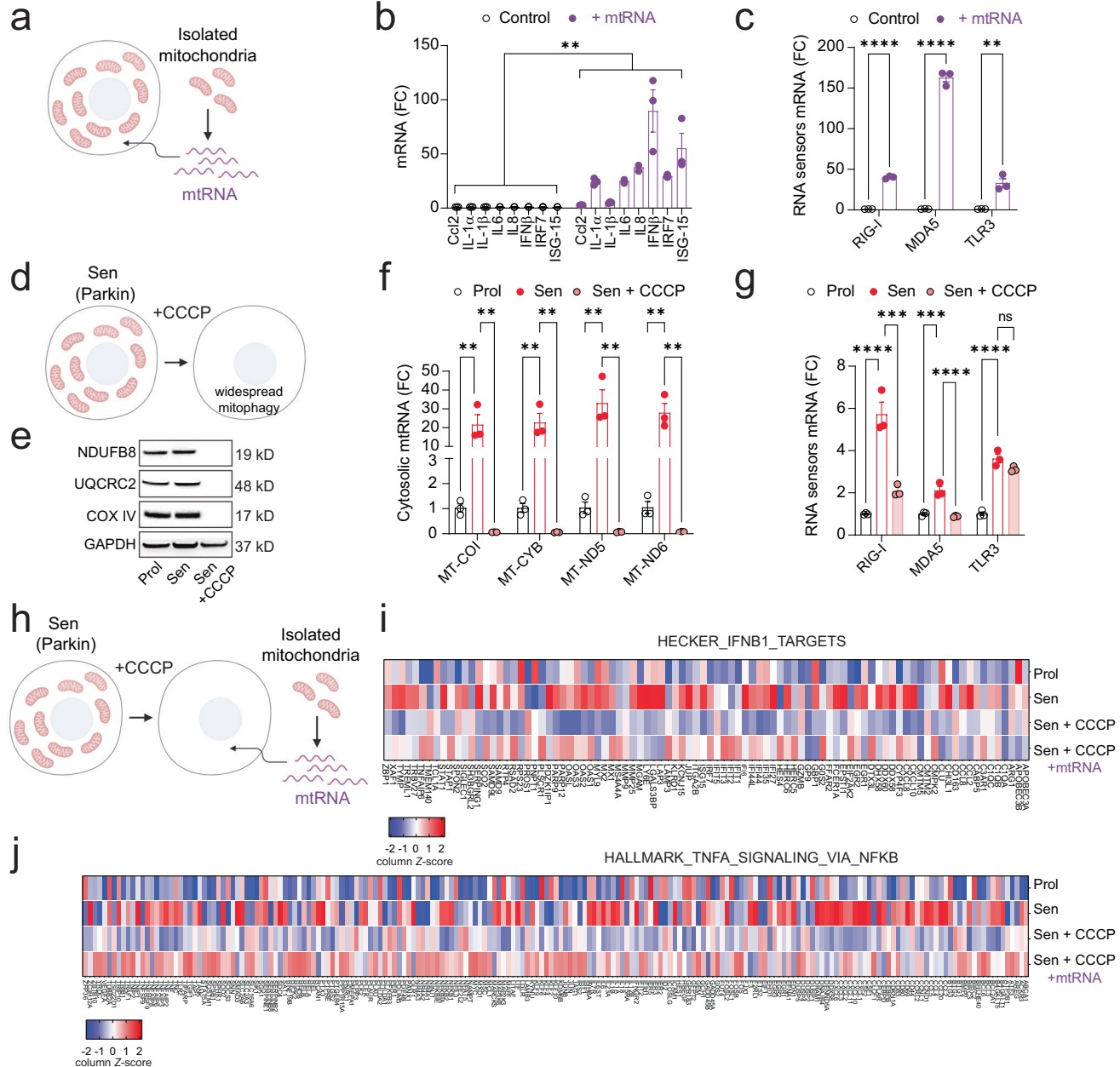

**Fig. 2 | Cytosolic mtRNA is a driver of the SASP. a** Scheme showing the isolation of mitochondrial RNA (mtRNA) and its subsequent transfection into cells. qPCR quantification of (**b**) SASP factors (*n* = 3 independent experiments) and (**c**) RNA sensors in proliferating MRC5 fibroblasts and following transfection with mtRNA. *n* = 3 independent experiments. **d** Scheme representing Parkin-mediated wide-spread mitophagy following CCCP treatment. **e** Western blot showing the expression levels of mitochondrial proteins NDUFB8, UQCRC2 and COX IV, demonstrating the absence of mitochondrial proteins following Parkin-mediated clearance. qPCR quantification of (**f**) cytosolic mtRNA genes MT-COI, MT-CYB, MT-ND5, MT-ND6 and (**g**) RNA sensors in Parkin-expressing IMR90 fibroblasts after widespread mitophagy. *n* = 3 independent experiments. **h** Experimental scheme showing senescent Parkin-expressing IMR90 fibroblasts being transfected with mtRNA following Parkin-mediated widespread mitophagy. Heatmaps showing

RNA-seq analysis of (**i**) IFNB1 target genes and (**j**) TNFA–NF-κB signaling pathway genes in proliferating, senescent, mitochondria-depleted senescent cells (Sen + CCCP), and mitochondria-depleted senescent cells transfected with mtRNA. Colors represent row Z-scores of gene expression. *n* = 3 (Prol and Sen) and *n* = 6 (Sen + CCCP and Sen + CCCP + mtRNA) independent experiments. Statistical significance was assessed by a two-tailed Nested *t*-test ($p$ = 0.0056) (**b**), two-sided Student's unpaired *t*-test ($p$ < 0.0001 for RIG-I and MDA5, $p$ = 0.0044 for TLR3) (**c**) and a one-way ANOVA followed by Tukey's multiple comparison test (**f**, **g**). **f** $p$ = 0.0078, $p$ = 0.0062, $p$ = 0.0039, $p$ = 0.0031, $p$ = 0.0037, $p$ = 0.0032, $p$ = 0.0014, $p$ = 0.0012; **g** $p$ < 0.0001, $p$ = 0.0001, $p$ = 0.0002, $p$ < 0.0001, $p$ < 0.0001, $p$ = 0.0932; (**b, c, f, g**) Data are mean ± s.e.m. (**a, d, h**) Created in BioRender. Victorelli, S. (2025) https://BioRender.com/0umlflk.

No change in expression of PCNA, p21 and p16 protein (Fig. 3d, e) or mRNA proliferation genes normally downregulated in senescent cells was observed (Fig. 3h, i), suggesting that RIG-I and MDA5 regulate the SASP, but not the senescence-associated cell-cycle arrest. Together, these data suggest that RIG-I and MDA5 recognize cytosolic mtRNA in senescent cells and play a role in SASP regulation.

## MAVS aggregation contributes to the SASP
Upon binding to RNA, MDA5 and RIG-I oligomerize and interact with the adaptor protein MAVS (Mitochondrial Antiviral Signaling protein). MAVS oligomerization is a nucleation process where multiple MAVS molecules come together to form a large, functional aggregate[28]. The aggregation of MAVS is essential for activation of

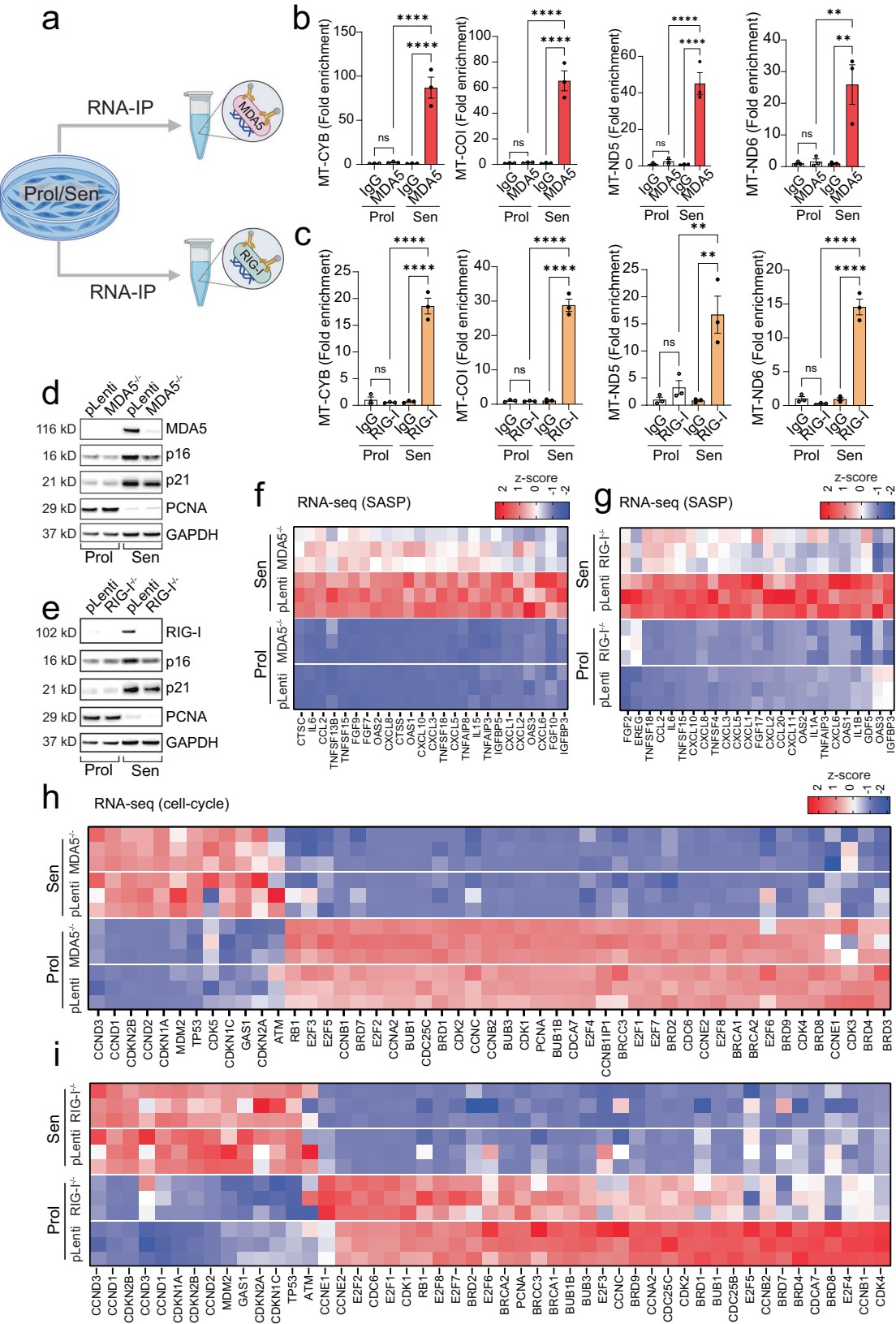

transcription factors such as IRF3 and NF-κB which play key roles in SASP regulation[29].

To investigate MAVS aggregation in senescence, we performed dual immunostaining and visualized the mitochondrial outer membrane protein TOM20 and MAVS using Airyscan microscopy. Our observations revealed that senescent cells exhibited distinct areas of irregularly shaped MAVS aggregates, which were not present in proliferating cells (Fig. 4a, b).

To further study the role of MAVS in senescence, we knocked it down using a pool of over 30 siRNAs to maximize knock down efficiency (Fig. 4c). RNA sequencing revealed that MAVS knockdown significantly reduced the expression of several SASP components during senescence,

**Fig. 3 | mtRNA engages MDA5 and RIG-I to regulate the SASP. a** Scheme representing experimental design for RNA immunoprecipitation (RNA-IP). Created in BioRender. Victorelli, S. (2025) https://BioRender.com/vkrj37g. qPCR quantification of mtRNA genes following RNA-IP of (**b**) MDA5 and (**c**) RIG-I in proliferating and senescent cells. $n = 3$ independent experiments. Western blot showing the successful deletion of (**d**) MDA5 and (**e**) RIG-I, as well as expression of senescence markers p16, p21 and PCNA in proliferating and senescent fibroblasts. Representative of $n = 3$ independent experiments. Column-clustered heatmap of SASP genes that are upregulated in senescence and significantly downregulated upon (**f**) MDA5 and (**g**) RIG-I deletion. The color intensity represents the column z-score. Column-clustered heatmap of cell-cycle related genes that are differentially expressed in senescent cells and not changed by **h** MDA5 and **i** RIG-I deletion. Color intensity represents the z-score. $n = 3$ independent experiments. Statistical significance was assessed by a one-way ANOVA followed by Tukey's multiple comparison test (**b, c**). **b** $p = 0.9988$, $p < 0.0001$, $p < 0.0001$, $p = 0.9998$, $p < 0.0001$, $p < 0.0001$, $p = 0.9853$, $p < 0.0001$, $p < 0.0001$, $p = 0.9993$, $p-0.0027$, $p = 0.0023$; (**c**) $p = 0.9763$, $p < 0.0001$, $p < 0.0001$, $p = 0.9998$, $p < 0.0001$, $p < 0.0001$, $p = 0.8213$, $p = 0.0038$, $p = 0.0012$, $p = 0.8314$, $p < 0.0001$, $p < 0.0001$. Data are mean ± s.e.m.

underscoring its critical role in this process (Fig. 4d). GO enrichment analysis of the top downregulated genes upon MAVS knockdown in senescent cells demonstrated a significant suppression of various inflammatory processes, notably the chronic inflammatory response and regulation of cytokine secretion (Fig. 4e). Additionally, similar to the deletion of MDA5 and RIG-I, MAVS knockdown in senescent cells resulted in a negative enrichment score for the SenMayo and SenSig gene panels (Fig. 4f). Importantly, MAVS knockdown did not significantly affect the expression of genes associated with cell proliferation, reinforcing its specific role in the modulation of the SASP (Fig. 4g).

Previous studies have established cGAS-STING as central regulators of the SASP[30], and we previously showed that cytosolic mtDNA contributes to their activation[17]. To determine the relative contribution of MAVS and cGAS-STING, we treated senescent cells with siRNA against MAVS, the STING inhibitor SN-011[31], or both (Supplementary Fig. 8a). STING inhibition markedly reduced expression of most SASP factors analyzed, while MAVS knockdown had milder effects. Notably, combined treatment did not produce additive suppression (Supplementary Fig. 8b). These findings suggest that MAVS and STING may converge on shared downstream pathways or act redundantly, with cGAS-STING playing the dominant role in SASP regulation.

## miMOMP is a driver of mtRNA cytosolic leakage in senescence

We have previously shown that the formation of BAX and BAK macropores, a process essential for mitochondrial outer membrane permeabilization (MOMP), in a subset of mitochondria in senescent cells leads to the leakage of mtDNA into the cytosol[17]. To investigate if the same mechanism causes the leakage of mtRNA during senescence, we utilized CRISPR-Cas9 gene editing to generate human fibroblasts deficient in both BAX and BAK (Fig. 5a, b).

Our findings reveal that the combined deletion of BAX and BAK significantly reduced the cytosolic release of mtRNA in senescent cells (Fig. 5c). This reduction was accompanied by decreased expression of RNA sensors RIG-I, MDA5, and TLR3 (Fig. 5d) and MAVS aggregation (Fig. 5e). Consistent with the role of BAX and BAK in mtRNA release and SASP activation, we confirmed that the deletion of BAX and BAK significantly reduced the expression of several SASP components (Fig. 5d). These results suggest that miMOMP, through the formation of BAX and BAK macropores, is a key driver of mtRNA cytosolic leakage in senescent cells, thereby contributing to the activation of the SASP. Work by the Garcia-Saez group demonstrated that BAX and BAK differ in their oligomerization dynamics and co-regulate apoptotic pore formation, thereby modulating the kinetics of mtDNA cytosolic release, with BAX forming larger, slower-assembling pores and BAK facilitating early BAX recruitment[32]. To explore whether these differences influence SASP activation, we performed siRNA-mediated knockdown of BAX, BAK, or both in senescent cells. The inhibition of BAX markedly reduced the expression of key SASP factors, including IL-6, IL-8, and IL-1β, while targeting BAK had minimal impact (Supplementary Fig. 9a–e). Combined knockdown showed a trend toward further reduction, but BAX depletion alone had the strongest impact. These results suggest a predominant role for BAX in SASP regulation. However, further studies are needed to determine whether distinct mitochondrial components are differentially released through BAX- versus BAK-dependent pores.

To further investigate whether BAX/BAK and MAVS act through shared or distinct transcriptional programs, we performed a comparative transcriptomic analysis of senescent cells lacking BAX and BAK versus those with MAVS knockdown. Analysis of the SenMayo gene set, enriched for SASP-related transcripts, revealed that 32 genes were consistently downregulated in both conditions (Fig. 5g). Remarkably, 30 of these genes were predicted NFKB1 targets, highlighting a shared downstream axis through which mitochondrial stress, via either miMOMP or RNA sensing, converges on NF-κB to drive SASP gene expression (Fig. 5h).

To investigate the role of BAX and BAK in senescence in vivo, we utilized Bax^fl/fl Bak^−/− mice and performed tail vein injections with AAV8-TBG-Cre virus to specifically delete BAX in the liver. It is known that cells in Bak^−/− mice can still undergo MOMP, while deletion of both Bax and Bak effectively blocks MOMP[33].

We subjected these mice to a high Fructose, Fat, and Cholesterol (FFC) diet, commonly used as a model of metabolic dysfunction–associated steatohepatitis (MASH), which has been shown to induce senescence in the liver[34] (Fig. 6a). As expected, FFC diet led to a marked increase in the senescence marker p21 protein expression (Fig. 6b, c) and elevated levels of telomere-associated DNA damage foci (TAF) (Fig. 6d), confirming the presence of senescent cells in the liver. To assess activation of RNA-sensing pathways, we performed a proximity ligation assay to detect co-localization between RIG-I and MAVS, a readout of MAVS activation, and found a significant increase in mice fed FFC diet (Fig. 6e, f). This was accompanied by elevated expression of RNA sensors, including Rig-I, Mda5, and Tlr3 (Fig. 6g). Consistent with an inflammatory response, we also observed increased mRNA levels of inflammatory cytokines (Fig. 6h) and markers of immune cell infiltration and activation, such as Cd45 and Cd68 (Fig. 6i). Together, these findings indicate that FFC diet induces senescence and activates RNA sensing and inflammatory pathways in the liver.

Supporting the hypothesis that Bax and Bak regulate the SASP via RNA sensor expression during diet-induced MASH, we found that hepatocyte-specific deletion of Bax and Bak (Fig. 6 j-l) significantly reduced the expression of RNA sensors Rig-I, Mda5, and Tlr3, as well as inflammatory factors, and markers of fibrosis (Fig. 6m, n & Supplementary Fig. 10a). The deletion of BAX and BAK did not affect body or liver weights and showed no impact on apoptosis markers (Supplementary Fig. 10b–e). This suggests that their influence on SASP and RNA sensor expression is independent of cell death. Hepatocyte-specific deletion of Bax and Bak significantly altered gene expression programs associated with inflammation (Fig. 6o), including a marked downregulation of commonly described SASP factors (Fig. 6p). Gene set enrichment analysis revealed negative enrichment scores for both RIG-I signaling and cytosolic DNA sensing pathways in Bax/Bak-deficient livers (Fig. 6q), suggesting that Bax and Bak deletion dampens not only RNA sensing but also DNA-driven inflammatory signaling. In addition, deletion of Bax and Bak markedly reduced RIG-I/MAVS co-localization, indicating decreased MAVS activation (Supplementary Fig. 10f). These findings further support a central role for miMOMP in activating both RNA- and DNA-dependent components of the SASP in vivo.

To further investigate the specific contribution of RNA signaling via MAVS to SASP activation in vivo, we used an AAV8-mediated

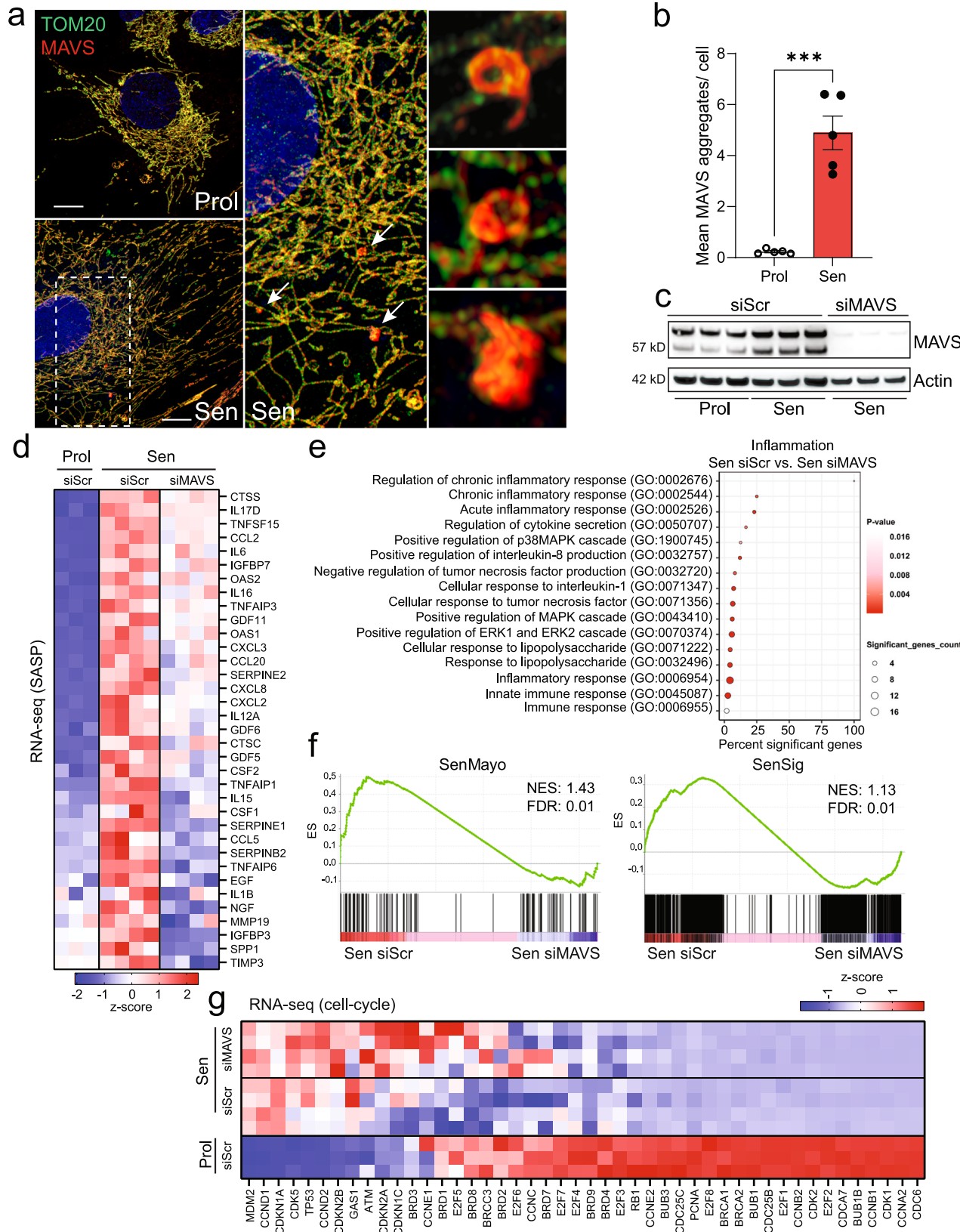

approach to deliver shRNA targeting MAVS specifically to hepatocytes in Western diet-fed mice (Supplementary Fig. 11a). Knockdown of MAVS led to a significant reduction in the expression of multiple inflammatory factors in the liver (Supplementary Fig. 11b), reinforcing the role of MAVS, and by extension RNA sensing, in promoting the SASP under MASH conditions. Notably, expression of the RNA sensors

RIG-I, MDA5, and TLR3 was also reduced (Supplementary Fig. 11c), suggesting a MAVS-dependent positive feedback loop that reinforces RNA sensor expression. Consistent with our in vitro findings, MAVS knockdown did not alter the expression of the senescence-associated cell cycle inhibitors p16 or p21 (Supplementary Fig. 11d), indicating that MAVS primarily regulates the inflammatory arm of the senescence

**Fig. 4 | Aggregation of MAVS is a feature of senescent cells and plays a role in SASP regulation. a** Representative super-resolution Airyscan microscopy image of TOM20 (green) and MAVS (red) in proliferating and senescent (IR) MRC5 fibroblasts showing MAVS aggregation in senescent cells. Arrows indicate MAVS aggregates that are amplified on the right. Scale bar=10 μm. **b** The mean number of MAVS aggregates observed in proliferating and senescent MRC5 human fibroblasts. *n* = 5 independent experiments. Statistical significance was assessed by a two-sided Student's unpaired t-test. p = 0.0001. Data are mean ± s.e.m. **c** Western blot showing successful small interfering RNA (siRNA) knockdown of MAVS in senescent MRC5 human fibroblasts. *n* = 3 independent experiments. **d** Column-clustered heatmap of SASP genes upregulated in senescent cells and significantly down-regulated upon knockdown of MAVS. Color intensity represents the z-score. *n* = 3-4 independent experiments. **e** Gene ontology (GO) term enrichment analysis showing pathways related to inflammation that are significantly altered between senescent control (Sen siScr; *n* = 4 independent experiments) and senescent cells lacking MAVS (Sen siMAVS; *n* = 4 independent experiments). **f** The GSEA plots for Sen siScr and Sen siMAVS show an enrichment of SenMayo and SenSign genes in senescent control cells. **g** Column-clustered heatmap of cell-cycle genes upregulated in senescent cells and not changed by MAVS siRNA knockdown. Color intensity represents the column z-score. *n* = 3–4 independent experiments.

program without affecting the growth arrest component. Supporting a reduction in SASP-mediated immune recruitment, MAVS knockdown also led to decreased infiltration of CD68⁺ macrophages in the liver (Supplementary Fig. 11e, f). Furthermore, MAVS expression positively correlated with serum levels of AST and ALT, with CD68+ infiltrates and with SASP factors Cxcl14, Cxcl1 and Oas1b (Supplementary Fig. 11h). Together, these findings establish MAVS as a driver of SASP-associated inflammation in MASH.

## Discussion

Mitochondrial dysfunction is a hallmark of cellular senescence[35] and a critical driver of the senescence-associated secretory phenotype (SASP)[15]. While the SASP has been attributed to the activation of DNA sensing pathways such as cGAS-STING[30] by either nuclear[36,37] or mitochondrial DNA[17], the role of mitochondrial RNA (mtRNA) in cellular senescence has not been investigated.

Our study provides new insights into the role of mtRNA as a driver of the SASP. Mitochondrial RNA shares structural similarities with bacterial RNA, such as unmethylated CpG motifs, which are recognized by pattern recognition receptors (PRRs) as pathogen-associated molecular patterns (PAMPs), thereby triggering an immune response[38]. Moreover, mtRNA differs from nuclear mRNA in that it lacks a 5' end methylated cap[39]. Under normal conditions, mtRNA is confined within mitochondria, and its release into the cytosol is abnormal. However, mtRNA can form double-stranded RNA (dsRNA) structures, and when present in the cytosol, dsRNA is typically associated with viral infections, leading the immune system to recognize it as a danger signal[19,40].

We show that mtRNA accumulates in the cytosol of senescent cells and activates RNA sensors RIG-I and MDA5, leading to MAVS aggregation and SASP induction. Inhibiting RIG-I, MDA5, or MAVS significantly reduced SASP gene expression. A recent study found that mtdsRNA, driven by PNPase or SUV3 knockdown, primarily localizes to stress granules and shows little co-localization with MDA5, suggesting transient interactions[41]. In contrast, our data in senescent cells indicate stable MDA5−mtRNA engagement: MDA5 immunoprecipitation followed by qPCR detected mtRNA binding in senescent but not proliferating cells. This difference may reflect variations in cell type, how mtdsRNA accumulates, or the timing at which MDA5−RNA interactions are captured.

We further show that this process depends on the formation of macropores by BAX and BAK in a subset of mitochondria, that allows mtRNA leakage. Loss of BAX and BAK reduces cytosolic mtRNA, downregulates RNA sensor expression, decreases RIG-I/MAVS co-localization, and suppresses the SASP both in vitro and in vivo. While previous studies showed that miMOMP inhibition can limit SASP activation during aging[17], we extend these findings to a model of MASH, a disease context characterized by increased senescence and inflammation. Recent studies suggest that voltage-dependent anion channel (VDAC) contributes to cytosolic leakage of mtDNA during senescence[42] and aging[43]. Whether VDAC also mediates the release of mtRNA, as shown here for BAX/BAK, remains unknown.

Our findings align with and extend those recently reported by the Serrano lab[44]. Both studies show that cytosolic mtRNA accumulates during senescence and that inhibition of RNA sensing pathways reduces the SASP. However, our study provides additional mechanistic and physiological insights. First, we show that mtRNA directly binds to the RNA sensors RIG-I and MDA-5 in senescent but not proliferating cells. Second, we provide transcriptome-wide data comparing the effects of RNA sensor inhibition (RIG-I, MDA5, MAVS, BAX, and BAK) as well as transfection of mtDNA or mtRNA into senescent cells lacking mitochondria, allowing direct comparisons of their downstream signaling programs. Our findings reveal that while mtDNA and mtRNA converge on shared inflammatory pathways, they also regulate distinct SASP gene modules. This suggests a layered and complex contribution of mitochondrial nucleic acids to SASP regulation. Finally, we validate our findings in vivo using a mouse model of MASH, showing that inhibition of BAX/BAK-mediated miMOMP and downstream MAVS signaling reduces SASP expression, and improves liver pathobiology.

Together, these findings support a broader model in which mitochondrial nucleic acids, both RNA and DNA, act as danger signals that trigger innate immune pathways in senescent cells. Future studies will be needed to clarify how different nucleic acid species interact or cooperate to shape the inflammatory landscape of senescent cells. Targeting these mitochondrial-derived signals may offer new opportunities for modulating inflammation in aging and age-related disease.

## Methods
### Cell culture and treatments
Human embryonic lung MRC5 fibroblasts and IMR90 fibroblasts were grown in Dulbecco's Modified Eagle's Medium (Sigma, D5796) supplemented with 10% heat-inactivated fetal bovine serum (FBS), 100 units/ml penicillin, 100 μg/ml streptomycin, and 2 mM L-glutamine and maintained at 37 °C with 5% CO₂. MRC5 fibroblasts were cultured in atmospheric oxygen conditions and IMR90 fibroblasts were cultured under low oxygen (3%) conditions.

HEK293T cells were used for lentiviral transduction and were cultured in DMEM as described above and further supplemented with 1% non-essential amino acids (Sigma, M7145), 500 μl/μl G418 antibiotic (Sigma, A1720) and 1 mM sodium pyruvate (Sigma, S8636).

Stress-induced senescence was achieved by exposing cells to X-ray irradiation at 10 Gy (MAFs) or 20 Gy (human fibroblasts). Replicative senescence was achieved by serially passaging cells until they reached their replicative potential and performed less than 0.5 population doublings for at least 4 weeks. For chemotherapy-induced senescence, cells were treated with either 250 nM of doxorubicin or 50 μM of etoposide for 24 hours and harvested at day 10 or day 8 post-treatment, respectively. Senescence was confirmed by presence of p16 and p21, and absence of proliferation markers Ki67 or EdU incorporation.

For inhibition of mitochondrial RNA polymerase (POLRMT), cells were exposed to 20 Gy X-ray irradiation and treated with 1 μM of the inhibitor IMT-1 (MedChem, HY-134539) for 10 days. Treatment was refreshed every other day.

For TLR3 inhibition, cells were exposed to 20 Gy X-ray irradiation and treated with 50 μM of TLR3/dsRNA complex inhibitor (Sigma,

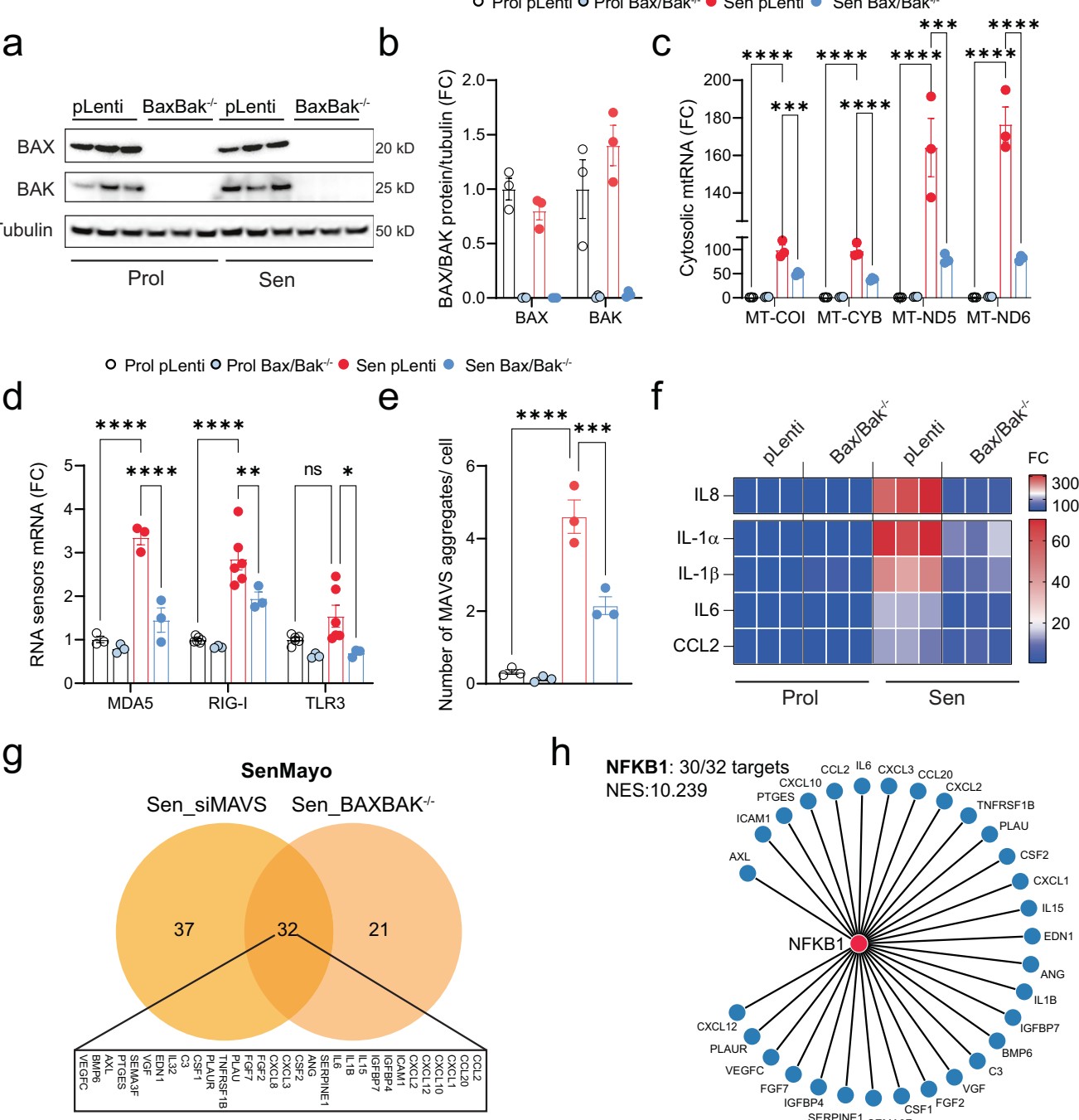

**Fig. 5 | BAX and BAK mediate the leakage of mtRNA into the cytosol of senescent cells.** Human MRC5 fibroblasts deficient in BAX and BAK (BAX$^{-/-}$ BAK$^{-/-}$) were generated using CRISPR-Cas9 gene editing. **a** Western blot confirming successful deletion of BAX and BAK in both proliferating and senescent MRC5 fibroblasts. $n = 3$ independent experiments. **b** Quantification of the Western blots shown in (**a**). **c** qPCR quantification of cytosolic mitochondrial RNA (mtRNA) transcripts in proliferating and senescent BAX$^{-/-}$ BAK$^{-/-}$ cells. $n = 3$ independent experiments. (**d**) mRNA expression of cytosolic RNA sensors in proliferating and senescent BAX$^{-/-}$ BAK$^{-/-}$ cells. (MDA5) $n = 3$, (RIG-I and TLR3) $n = 6$ (Prol pLenti, Sen pLenti) $n = 3$ (Prol Bax/Bak$^{-/-}$ and Sen Bax/Bak$^{-}$) independent experiments. **e** Quantification of MAVS aggregates *per* cell based on immunofluorescence staining for TOM20 and MAVS in proliferating and senescent BAX$^{-/-}$ BAK$^{-/-}$ cells. $n = 3$ independent experiments.

**f** mRNA levels of SASP factors in proliferating and senescent BAX$^{-/-}$ BAK$^{-/-}$ cells. $n = 3$ independent experiments. **g** Venn diagram depicting the overlap of commonly downregulated SenMayo genes between senescent cells lacking MAVS (Sen_siMAVS) and deficient for BAX/BAK (Sen_BAXBAK$^{-/-}$), based on RNA sequencing. **h** Using iRegulon, the transcription factor NF-κB1 was found to control 30 of the 32 overlapping target genes from (**g**), indicating it is the most likely key regulator (NES 10.239) for these SASP factors. Data are mean ± s.e.m. Statistical significance was assessed by a one-way ANOVA followed by Tukey's multiple comparison test. **b** $p < 0.0001$, $p = 0.0001$, $p = 0.3181$, $p = 0.0162$, $p = 0.0021$, $p = 0.5401$; **c** $p < 0.0001$, 0.0006, $p < 0.0001$, $p < 0.0001$, $p < 0.0001$, $p = 0.0004$, $p < 0.0001$, $p < 0.0001$; **d** $p < 0.0001$, $p < 0.0001$, $p < 0.0001$, $p = 0.0058$, $p = 0.0652$, $p = 0.0109$; **e** $p < 0.0001$, $p = 0.0008$.

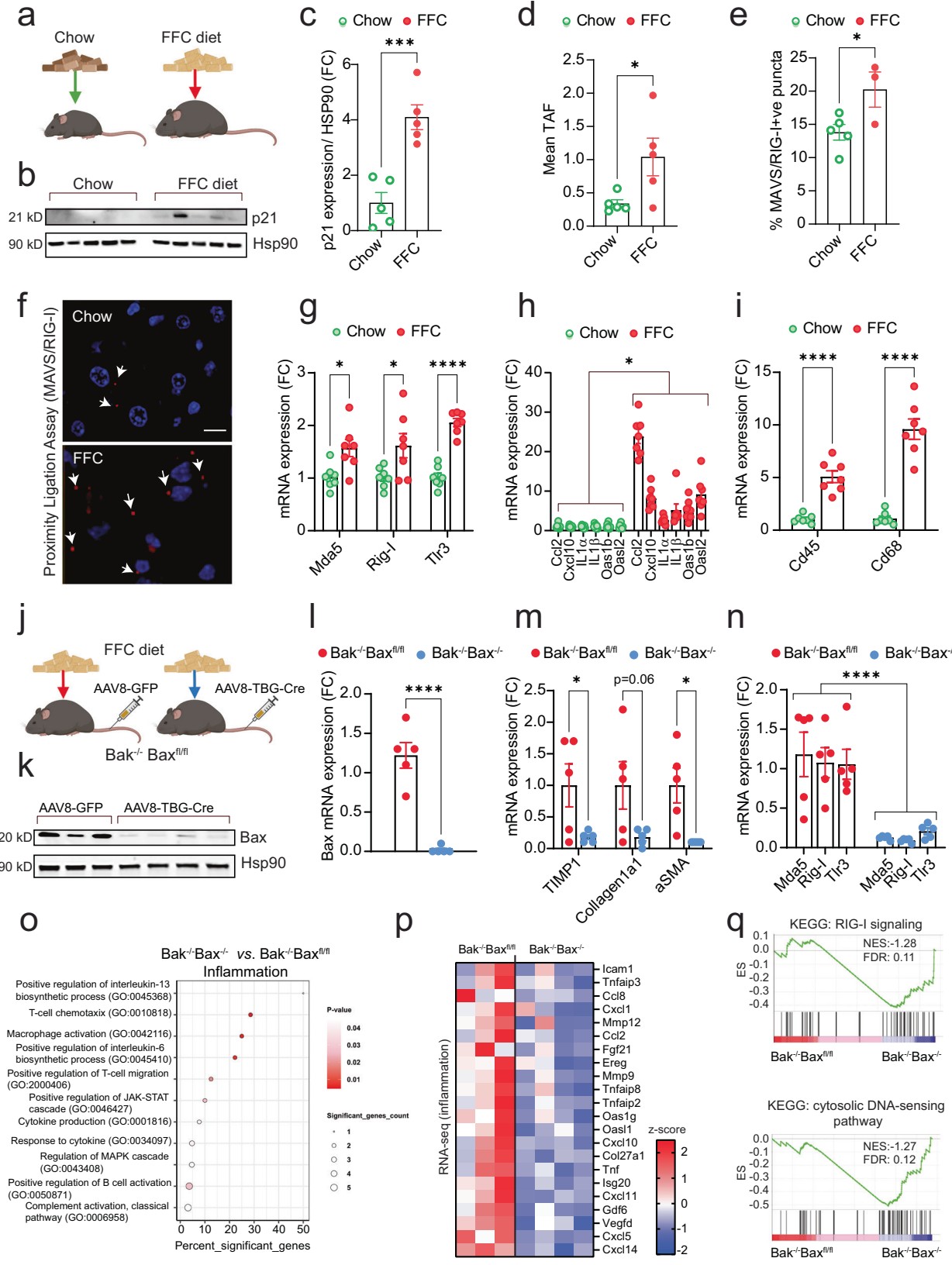

614310) for 10 days. Treatment was refreshed every day for the duration of the experiment.

For STING inhibition, IMR90 fibroblasts were exposed to 20 Gy X-ray irradiation and treated with 10 μM SN-011(Cayman Chemical, NC2044999) for 10 days. Treatment was refreshed every other day.

**Parkin-mediated mitochondria clearance**

Parkin-mediated widespread mitochondrial clearance was performed as in ref. 14,15. Briefly, proliferating, or irradiated Parkin-expressing IMR90 fibroblasts were treated with 12.5 μM CCCP (Sigma, C2759) (3 days after irradiation) for 48 hours (CCCP was refreshed every

**Fig. 6 | Hepatocyte-specific deletion of BAX/BAK reduces expression of RNA sensors and inflammation in the liver during MASH. a** Schematic of experimental design: wild-type mice (10–12 weeks old) were fed either standard chow or a high-fat, fructose, and cholesterol (FFC) diet for 24 weeks to induce metabolic-associated steatohepatitis (MASH). **b** Western blot showing levels of the senescence marker p21 in liver tissue from chow- and FFC-fed mice. **c** Quantification of p21 protein levels from the Western blot shown in (**b**). $n = 5$ mice per group. **d** Quantification of telomere-associated DNA damage foci (TAF) in liver sections from chow- and FFC-fed mice, shown as mean TAF *per* nucleus. $n = 5$ mice per group. **e** Quantification of the percentage of MAVS/RIG-I double-positive puncta per field in liver sections from chow- and FFC-fed mice, measured by Proximity Ligation Assay (PLA). $n = 5$ chow- and 3 FFC-fed mice. **f** Representative PLA images. Scale bar = 30 μm. mRNA expression levels of **g** cytosolic RNA sensors, **h** SASP-related factors and **i** immune cell markers, Cd45 and Cd68 in the livers from chow and FFC-fed mice. $n = 7$ mice per group. **j** Experimental setup for hepatocyte-specific Bak and Bax deletion. Bak$^{-/-}$ Bax$^{fl/fl}$ mice (10–12 weeks old) were fed an FFC diet for 16 weeks to induce MASH. Mice were then injected with either AAV8-GFP (control) or AAV8-TBG-Cre virus via the tail vein to induce deletion of Bax specifically in hepatocytes. **k** Western blot showing successful deletion of Bax in the liver following AAV8-TBG-Cre injection. $n = 3$-4 mice per group. qPCR quantification of **l** Bax, **m** markers of fibrosis and **n** RNA sensors in livers of FFC-fed Bak$^{-/-}$ Bax$^{fl/fl}$ and Bak$^{-/-}$ Bax$^{-/-}$ mice. $n = 5$ mice per group. **o** Gene ontology (GO) term enrichment analysis showing pathways related to inflammation that are significantly altered between livers from FFC-fed Bak$^{-/-}$ Bax$^{fl/fl}$ ($n = 3$) and Bak$^{-/-}$ Bax$^{-/-}$ ($n = 4$) mice. **p** Column-clustered heatmap of significantly downregulated inflammation-related genes in the livers of FFC-fed mice lacking both Bax and Bak. Color intensity reflects column Z-score of gene expression. **q** Gene set enrichment analysis (GSEA) showing negative enrichment scores for both RIG-I–like receptor signaling and cytosolic DNA sensing pathways in Bak$^{-/-}$ Bax$^{-/-}$ livers. Data are mean ± s.e.m. Statistical significance was assessed by a two-sided Student's unpaired *t*-test (**c**, **d**, **g**, **i**, **l**, **m**) and a Nested *t*-test (**h**, **n**). **c** $p = 0.0007$; **d** $0.0413$; **e** $p = 0.0416$; **g** $p = 0.0114$, $p = 0.0284$, $p < 0.0001$; **h** $p = 0.0287$; **i** $p < 0.0001$, $p < 0.0001$; **l** $p < 0.0001$; **m** $p = 0.0436$, $p = 0.06$, $p = 0.0118$; (**n**) $p < 0.0001$. **a**, **j** Created in BioRender. Victorelli, S. (2025) https://BioRender.com/tdsqpjz.

---

24 hours). Mitochondria-depleted cells were then transfected with isolated mitochondrial RNA (as described below) at 8 days post-irradiation and harvested at 10 days after irradiation.

## Mitochondrial RNA transfection
Proliferating MRC5 fibroblasts and Parkin-expressing IMR90 fibroblasts were transfected with 2.5 μg of isolated mitochondrial RNA using Lipofectamine MessengerMax Reagent (Invitrogen, LMRNA003), according to the manufacturer's instructions. Cells were incubated with mtRNA for 48 hours and then harvested for analysis.

## Subcellular fractionation
For cytosolic fraction analysis, subcellular fractionation was performed using Subcellular Protein Fractionation Kit for Cultured Cells (Thermo Fisher, 78840), according to manufacturer's instructions.

## Mitochondrial isolation and mitochondrial RNA extraction
For the mitochondrial-enriched fraction, followed by a rinse in ice-cold PBS, cells were collected by scraping the flask with 5 ml of ice-cold PBS. Cells were centrifuged at 800 $g$ for 5 min at 4 °C and resuspended in mitochondrial isolation solution (MIS) (20 mM HEPES-KOH pH 7, 220 mM mannitol, 70 mM sucrose, 1 mM EDTA, 0.5 mM PMSF, 2 mM DTT). The samples were transferred to a glass homogenizer, and cells were broken open using 60 strokes. The homogenate was centrifuged at 800 $g$ for 5 min at 4 °C. The supernatant was further centrifuged at 800 $g$ for 5 min at 4 °C. An aliquot of the supernatant was collected and stored as the whole-cell extract. The remaining was centrifuged at 16,100 $g$ for 10 min at 4 °C. The supernatant was collected as the cytosolic fraction. The pellet containing mitochondria was resuspended in 1 ml of MIS and centrifuged again at 16,100 $g$ for 10 min at 4 °C. This step was repeated, and the resulting pellet was resuspended in 100 μl of MIS. For mitochondrial RNA extraction, the mitochondrial pellet was centrifuged at 16,100 $g$ for 10 min at 4 °C, and RNA extraction was performed using the QIAGEN RNeasy Mini Kit (Qiagen, 74106) according to manufacturer's instructions.

## CRISPR/CAS9-based genome editing
The following plasmids were used: LentiCRISPR v2 hBAK (Addgene, 129579), LentiCRISPR v2 hBAX (Addgene, 129580), LentiCRISPR v2-puro (Addgene, 52961), pLV[CRISPR]-hCas9:T2A:Puro-U6 > hDDX58[gRNA#3708] (VectorBuilder, VB900079-3851tta) and pLV[CRISPR]-hCas9:T2A:Puro-U6 > hIFIH1[gRNA#141] (VectorBuilder, VB900098-8309keh).

For lentiviral transduction, HEK293FT cells were transfected with the plasmids above together with the packaging and envelope plasmids VSVG and Gag-Pol (Sigma) using Lipofectamine 3000 (Invitrogen, L3000015) according to the manufacturer's instructions. Two days later, the supernatant from the transfected HEK293FT cells containing viral particles was filtered using a 0.45 μm pore PVDF filter, mixed with 10 μg/ml polybrene and used to infect the cells of interest. Following infection, cells were selected for successful CRISPR/Cas9 deletion by using 1 μg/ml Puromycin.

## siRNA gene knockdown
MRC5 fibroblasts were transiently transfected with 30 nM of a pool of siRNAs against MAVS (Galen Molecular, si-G050-57506), Bax (Galen Molecular, si-G050-581), Bak (Galen Molecular, si-G050-578) or negative control (Galen Molecular, si-C020). Cells were transfected using DharmaFECT 2 transfection reagent (Thermo Fisher, T-2002-03) following manufacturer's instructions. For senescent cells, cells were transfected one day prior to X-ray irradiation and then again on day 7 post-irradiation. Cells were harvested at day 10 following irradiation for analysis.

IMR90 fibroblasts were transfected with 30 nM of a pool of siRNAs against MAVS (Galen Molecular, si-G050-57506) or negative control (Galen Molecular, si-C020), as above. Cells were transfected one day prior to X-ray irradiation and harvested at day 10 following irradiation for analysis.

## qPCR
RNA was extracted using the RNAeasy Mini Kit (Qiagen, 74106) according to the manufacturer's instructions. Complementary DNAs were synthesized using the High-Capacity cDNA Reverse Transcription Kit (Thermo Fisher Scientific, 4368814) according to the manufacturer's instructions. qPCR was performed using ToughMix Perfecta (PerfeCTa qPCR ToughMix, QuantaBio, 95112-250) using the CFX96TM Real-Time System (Bio-Rad). mRNA levels were calculated using the 2 − ΔΔCt method and normalized to a housekeeping gene. DNA was quantified using the Nanodrop and stored at −20 °C. The primers used are listed in Supplementary Table 1.

## Western blotting
Cells were lysed in lysis buffer (150 mM NaCl, 1% NP40, 0.5% sodium deoxycholate, 0.1% SDS, 50 mM Tris pH 7.4, 1× phosphatase and protease inhibitors cocktail in $H_2O$) and the protein concentration was determined using the Bio-Rad protein assay (Bio-Rad, reagent A, 500-0113; reagent B, 500-0114; reagent C, 500-0115). Equal amounts of protein (at least 15 μg) from each sample were resolved on Tris-glycine gels and samples were then blotted onto a 0.45 μm polyvinylidene difluoride (PVDF) membrane (Millipore) using Trans-Blot SD Semi-Dry Transfer Cells (Bio-Rad). Membranes were blocked with PBS-Tween blocking buffer (5% milk powder, 0.05% Tween-20 in PBS) and then incubated with primary antibodies at 4 °C overnight (a list of the antibodies used is provided in Supplementary Table 2). After washes in

distilled water, the membranes were incubated with a peroxidase-conjugated secondary antibody for 1 h at room temperature. The membranes were then incubated with either Clarity ECL Western Blot Substrate (Bio-Ras, 170–5060) or the KwikQuant Western blot detection kit (Kindle Bioscience, R1100) according to manufacturer's instructions, and visualized using iBright 1500 (Invitrogen). The antibodies used are listed in Supplementary Table 2.

## RNA Immunoprecipitation (RIP) Assay

RNA immunoprecipitation (RIP) assay was performed using the Magna RIP RNA-Binding Protein Immunoprecipitation Kit (Merck Millipore, MA, USA) according to manufacturer's directions. Briefly, 8 ×106 MRC5 cells (proliferating and senescent) were scraped from T175 flasks, centrifuged at 500 g for 5 minutes, and washed in cold 1X PBS. The pellets were then resuspended in 200 μl of RIPA cell lysis buffer. The Magnetic Beads Protein A/G (CS203178; Merck Millipore, MA, USA) were suspended in RIPA cell lysis buffer, bound to a magnetic separator and washed with RIPA wash buffer and again bound to the magnetic separator with the supernatant being removed. The magnetic beads were resuspended in 100 μl of RIPA wash buffer. 5 μg of RIG-I (20566-1-AP; Proteintech, IL, USA) or MDA5 (21775-1-AP; Proteintech, IL, USA) or rabbit IgG antibody were preincubated with beads, washed, and resuspended in 900 μl of RIP Immunoprecipitation buffer including EDTA and RNAse inhibitor. RIP lysate was centrifuged at 18,800 g for 10 minutes at 4 °C. 100 μL of the supernatant was added to each beads-antibody complex in RIP Immunoprecipitation Buffer and incubated overnight rotating at 4 °C. The following day, the lysate-beads-antibody complex was washed 5X in cold RIP wash buffer. Following the final wash step, the lysate-beads-antibody complex was incubated in the Proteinase K buffer at 55 °C for 30 minutes. The tubes were placed on the magnetic separator and the supernatant was removed followed by addition of 250 μl of RIP buffer. RNA was subsequently isolated using TRIzol™ Reagent (Invitrogen, Carlsbad, CA) according to manufacturer's directions. RNA was reverse transcribed using SuperScript III First-Strand synthesis (Invitrogen, Carlsbad, CA) and qPCR for mtRNA transcripts was performed.

## RNA-sequencing

Sequencing libraries were made from poly-A RNA, as recommended by Illumina, and sequenced using either an Illumina GAIIX or a NextSeq 500 sequencer. RNA-seq paired-end reads were assessed for quality using the 'FastQC' algorithm, then aligned to the human genome using the splice-aware aligner STAR with a two-pass alignment pipeline. Reference splice junctions were provided by a reference transcriptome from the Gencode GRCh38 (hg38) build. BigWig files were generated using DeepTools. Raw read counts per gene were calculated using htseq-count. The read count matrix was then used for differential expression analysis with the linear modeling tool DESeq2. Significantly changing expression was defined as an FDR-corrected $p$-value ≤ 0.05. FPKM (Fragments Per Kilobase of transcript per Million mapped reads) values were generated using Cufflinks. Gene ontology analysis was performed using Gene Set Enrichment Analysis (GSEA) and Ingenuity® Pathway Analysis (IPA) software. Transcription factor analysis was created iRegulon in Cytoscape. A gene-based motif collection (1120 ChIP-seq tracks, regulatory region 20 kb centered around TSS, ROC threshold for AUC: 0.03, Rank threshold 5.000) was used for the analysis.

## Immunocytochemistry

Cells grown on coverslips were fixed using 3.7% paraformaldehyde in DMEM for 10 minutes. Cells were washed in PBS and then permeabilized in 0.1% Triton X-100 in PBS for 10 minutes at room temperature. Following blocking in 1% bovine serum albumin (BSA) in PBS-Tween20 for 1 hour, cells were then with primary antibody overnight at 4 °C (antibodies diluted in 1% BSA/PBS). The following day, cells were

washed three times in PBS-Tween20, and secondary antibody was applied and incubated for 45 minutes at room temperature. Coverslips were mounted onto glass microscope slides with ProLong Gold Antifade Mountant with DAPI (Invitrogen).

For MAVS, cells were fixed in 0.05% glutaraldehyde in 4% paraformaldehyde for 10 minutes.

The antibodies used are listed in Supplementary Table 3.

## Terminal deoxynucleotidyl transferase-mediated deoxyuridine triphosphate nick-end labeling (TUNEL) assay

Apoptosis in tissue was measured by the fluorescent terminal deoxynucleotidyl transferase-mediated deoxyuridine triphosphate nick-end labeling (TUNEL) assay on frozen liver sections. Liver tissue samples were cryopreserved in optimum cutting temperature compound (Takeda) immediately after harvest. Tissue sections were cut at 7 μm on a cryomicrotome (Leica) and stored at −80 °C before use. The TUNEL assay was then performed using the manufacturer's protocol (#11684795910, Roche). Apoptotic cells were quantified by counting TUNEL-positive nuclei per high-power field in 20 random microscopic fields using a fluorescent microscope (Eclipse 80i; Nikon) in a blinded manner.

## Immuno-FISH

FFPE tissue sections were deparaffinized in 100% Histoclear and hydrated through a graded ethanol series of 100, 90 and 70% ethanol (twice for 5 minutes each), and washed twice for 5 minutes in distilled water. Antigen retrieval was performed by placing sections in 0.01 M citrate buffer (pH 6.0) and heating it until boiling for 10 minutes. Sections were allowed to cool down to room temperature and then washed in distilled water for 5 minutes. Blocking was then performed using normal goat serum (1:60) in BSA/PBS for 30 minutes followed by overnight incubation with rabbit monoclonal anti-γH2AX antibody (1:400, 9718; Cell Signaling) at 4 °C. Following three PBS washes, tissues were incubated with a goat anti-rabbit biotinylated secondary antibody (1:200, PK-6101; Vector Labs) for 30 minutes at room temperature. Sections were then washed in PBS three times and incubated with fluorescein avidin DCS (1:500, A-2011; Vector Labs) for 30 minutes at room temperature. Tissues were then washed in PBS three times and incubated in 4% paraformaldehyde in PBS for 20 minutes for cross-linking. Following three PBS washes, sections were dehydrated in graded cold ethanol solutions (70, 90, 100%) for three minutes each and were then allowed to air-dry. Next, 10 μl of PNA hybridization mix (70% deionized formamide (Sigma), 20 mM MgCl2, 1 M Tris pH 7.2, 5% blocking reagent (Roche) containing 2.5 μg/m Cy-3-labeled telomere-specific (CCCTAA) peptide nucleic acid probe (PANAGENE)) was added to sections and denaturation was allowed to occur for 10 minutes at 80 °C. Sections were then incubated in PNA hybridization mix for 2 hours at room temperature in the dark to allow hybridization to occur. Tissues were washed in 70% formamide in 2X SSC for 10 minutes, followed by one wash in 2X SSC for minutes and a PBS wash for 10 minutes. Tissues were mounted using ProLong Gold Antifade Mountant with DAPI (Invitrogen) and imaged using in-depth Z stacking (a minimum of 40 optical slices with 63x objective).

## Immunohistochemistry

For CD68 immunohistochemical staining, paraffin-embedded tissue sections were dewaxed, rehydrated, and subjected to antigen retrieval using PT-Link buffer (pH 9). Endogenous peroxidase activity was blocked by incubating the sections with 3% hydrogen peroxide for 10 minutes. Slides were then incubated overnight at room temperature with the primary antibody against CD68 (Abcam, ab12512; 1:100 dilution), followed by a 30-minute incubation with a secondary antibody (EnVision system; DAKO) at room temperature. Visualization was achieved using a DAB chromogen for 1 minute, and sections were counterstained with Mayer's hematoxylin for 10 minutes. Finally,

samples were dehydrated and mounted. For image analysis, up to four representative microphotographs per animal were captured using an Olympus BX51 microscope equipped with a DP70 digital camera. CD68-positive areas were quantified using ImageJ software (version 1.52p).

## Proximity Ligation Assay (PLA)

Proximity ligation assay (PLA) was performed as an approach to determine protein-protein interactions in vivo. Here, Duolink In Situ Detection Reagents Red (Sigma, DUO92008) was used to detect interactions between RIG-I and MAVS. The PLA was performed according to manufacturer's directions using mouse formalin-fixed paraffin-embedded (FFPE) sections. The following primary antibodies were used: rabbit RIG-I (1:100) (20566-1-AP, Proteintech) and mouse monoclonal MAVS (1:100) (clone E3, Santa Cruz Biotechnology). After washing the slides, mouse-minus and rabbit-plus probes were incubated to the slides for 60 min at 37 °C. Next the ligase was diluted 1:40 and incubated for 30 min at 37 °C. Again, washes were performed; polymerase was diluted 1:80 and incubated for 100 min at 37 °C. The slides were washed 2 × 10 min in 1× wash buffer B and washed for 1 min with 0.01× wash buffer B prior to mounting the samples with Duolink in situ mounting medium with DAPI. The slides were analyzed using a Zeiss 780 laser scanning confocal microscope.

## Microscopy

Imaging was performed using widefield microscope (DMi8 Leica) and super-resolution microscope (confocal microscopy using the AiryScan type detector LSM980 Zeiss AiryScan).

Analysis of cytosolic mitochondrial RNA and MAVS aggregation was performed using ImageJ.

## Mouse models

All animal experiments were performed according to protocols approved by the Institutional Animal Care and Use Committee (IACUC) at Mayo Clinic. Mice were kept at the Mayo Clinic Rochester animal facility with a 12-hour light-dark cycle. All mice were placed on a diet at 10–12 weeks of age. The chow diet was a standard rodent diet (PicoLab 5053, LabDiet) with tap water. FFC diet contains high fat (40% calories) and high cholesterol (0.2%) (#AIN-76A Western Diet; TestDiet), with fructose (18.9 g/L) and glucose (23.1 g/L) added to the drinking water (total sugar 42 g/L) as previously described (PMID: 37267252). For characterizing senescence and expression of RNA sensors in the liver, C57Bl6/J male mice (purchased from the Jackson Laboratory) were placed on a chow or FFC diet for 24 weeks. For Bak and Bax studies, 10–12 week-old Bak$^{-/-}$ Bax$^{fl/fl}$ male mice (obtained from the Jackson Laboratory, strain #006329) were injected via tail vein with $1.5 × 10^{11}$ GC/mouse of either AAV8-TBG-Cre (for hepatocyte-specific deletion) or with control virus (AAV8-GFP), both obtained from Vector Builder. Two weeks post-injection, mice were placed into a thermoneutral caging system (Solace Zone) and were kept on the FFC diet for 16 weeks. At the end of the feeding period, mice were sacrificed under general anesthesia induced by a combination of xylazine and ketamine. The liver was harvested and processed for further examination.

Experiments involving Alfp-Cre mice were performed in accordance with standard protocols approved by the Animal Ethics Committee of the University of Santiago de Compostela. All animals received the highest standards of humane care throughout the study. Alfp-Cre mice, which express the Cre recombinase open reading frame (ORF) under the control of mouse albumin regulatory elements and α-fetoprotein enhancers, mimicking the endogenous genomic organization of the albumin gene, were obtained from The Jackson Laboratory. Mice were housed in temperature-controlled rooms (22 °C) under a 12:12 h light/dark cycle. At 8 weeks of age, animals were provided ad libitum access to either a Western diet (TD88137; 42% kJ from fat, 15% from protein, 43% from carbohydrates, and 1.25% cholesterol; Ssniff) or a matched control chow diet (CD88137; 13% kJ from fat, 18% from protein, and 69% from carbohydrates; Ssniff) for a period of 9 weeks.

For intravenous injections, mice were placed in a specialized restrainer (Tailveiner, TV-150; Bioseb). Injections were performed via the lateral tail vein using a 27 G × 3/8" (0.40 mm × 10 mm) needle. Mice received 100 μL of either AAV8-loxP-shMAVS ($1 × 10^9$ TU/mL) or AAV8-GFP ($1 × 10^9$ TU/mL), both diluted in sterile saline. In the cohort fed a Western diet for 9 weeks, AAV8-loxP-shMAVS was administered during the first week of dietary intervention as described in ref. [45]. At the end of the experimental period, mice were euthanized by decapitation. Tissues were immediately harvested, snap-frozen in dry ice, and stored at −80 °C until further analysis.

## Ethics statement

All animal experiments were performed according to protocols approved by the Institutional Animal Care and Use Committee (IACUC) at Mayo Clinic.

## Reporting summary

Further information on research design is available in the Nature Portfolio Reporting Summary linked to this article.

## Data availability

The RNA-seq datasets generated and analyzed during the current study are available in the GEO repository GSE306839 and GSE306957. Source data are provided with this paper.

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

## Acknowledgements

This work was funded by NIH grants R01AG068048 (JFP); R01AG82708 (JFP); UH3CA268103 (JFP); P01 AG062413 (JFP); R01DK057993 (NFL, SPO); R01DK130884 (PH), Robert and Arlene Kogod Center on Aging Career Development Award (SV, HM), Center for Cell Signaling in Gastroenterology (P30 DK084567) for both Pilot and Feasibility Award (SV) and Optical Microscopy Core, The Glenn Foundation For Medical Research (JFP); R01 AG068182 (DJ), Hevolution/AFAR (DJ). Research reported in this publication was supported by the National Institute of Diabetes and Digestive and Kidney Diseases of the National Institutes of Health under Award Number P30DK084567. The content is solely the responsibility of the authors and does not necessarily represent the official views of the National Institutes of Health. This publication was supported by NIGMS Grant Number 5T32GM145408 from the National Institute of General Medical Sciences (NIGMS). Its contents are solely the responsibility of the authors and do not necessarily represent the official views of the NIH.

## Author contributions

S.V. and M.E. performed most of the experiments. S.V. performed and evaluated individual experiments. H.M., S.H.W., S.E., G.L., N.P., N.H., E.Y.L., A.C.F., E.N., P.S., S.P.O., O.M., L.V.P., H.S.K.L. contributed to individual experiments. Y.H. and D.S. contributed to the analysis of RNA-sequencing datasets. R.N., D.J., N.F.L., P.H. supervised individual experiments. J.F.P. and S.V. designed and supervised the study; J.F.P. and S.V. wrote the manuscript with contributions from all the authors.

## Competing interests

The authors declare no competing interests.
