## [Transparent Peer Review file · Nature Communications]

Mitochondrial RNA cytosolic leakage drives the SASP

Corresponding Author: Dr Stella Victorelli

Version 0:

Reviewer comments:

Reviewer #1

(Remarks to the Author)

The work phenocopies studies by this group showing the role of mtDNA release for cGAS and STING activation as a driver of the SASP, and is well in line with a recently released paper by the Serrano group, also published in Nature Communications, early last week?! PMID: 39191740

The authors convincingly show that mitochondria-derived dsRNA are found at increased levels in the cytoplasm of senescent cells. This can activate cytosolic RNA sensors (known) to drive the senescence associated secretory phenotype, but is unrelated to the senescence induced cell cycle block via p53, p16, p21. They go on and show that loss of cytosolic RNA sensors RIGI and MDA5 or MAVS (but not TLR3) are important for the inflammatory Nf-kB signature found in senescent cells. Release of mtdsRNA requires the BCL2 family proteins BAX and BAK (based on using double-deficient cells as reference; a finding reported before). Finally, they show in a model of western-diet induced liver ageing, dsRNA sensors are increased and their expression as well as the secretory senescence phenotypes are reduced on a BAX/BAK mutant background.

The study is convincing, the experiments well done. I have only few concerns.

1) the authors show that both mtDNA and mtdsRNA can drive the SASP and the data looks very clean in both studies. However, one would expect some redundancy. Does combined loss of MAVS + cGAS (or STING) act in an additive manner - as one would expect, or are they fully redundant? This can be tested in one of the model systems by assessing the impact of combined vs. single deletion of MAVS and or cGAS (or STING)

2) Work by the Garcia-Saez group suggests that BAX and BAK influence each other, coordinating the growth of the "apoptotic pore" with Bax forming larger structures compared to BAK - is there any reason to believe that mtdsRNA vs. mtDNA may be released preferentially through one or the other, or are they fully redundant (discuss / maybe test Bax vs BAK single KO cells in one of the paradigms (e.g. cytokine mRNA induction)

3) The MASH data is correlative at best, as it does not show that diet-induced SASP phenotypes are driven by any of the RNA sensors or MAVS. This is my biggest worry, as it actually only shows dsRNA sensors are up in MASH mice, and BAX/BAK DKO mice show reduced damage/SASP. It could equally well be cGAS/STING mediated (expression can easily be elevated, too), and both may contribute to the phenotype (or lack thereof). So, I suggest to assess levels of cGAS/STING and signalling components downstream (TBK1 and IRF3) in parallel and adapt/discuss findings depending on outcome. I do not ask the authors to repeat these experiments using siRNAs against any of these components at that stage (even though it would be the relevant controls)

4) the authors may also discuss a recent study showing that mtdsRNA expression levels correlate with the proliferative state in fibroblasts and are reported to be very low in quiescent cells, while high upon their immortalization/transformation (PMID: 39209534). Another study misses to detect colocalization of MDA5 with mtdsRNA in the cytosol (PMID: 38955468) These observations don't add up with what is shown here and should be discussed, along with the role of VDACs in this process, in light of their own observations.

Reviewer #2

(Remarks to the Author)

The authors report a role of mtRNA on the SASP during senescence. They identify the release of mtRNA into the cytosol in senescent cells, which they associate with an increase in MDA5 and RIG-I expression and the activation of MAVS. They also show that cytosolic mtRNA can induce a SASP response, independent of other mitochondrial components. In line with the previous reported role of BAX and BAK on the release of mtDNA, they report that mtRNA release in senescence also depends on these proteins. And they show that BAX and BAK contribute to the SASP in the liver during MASH.

On the one hand, the results are kind of expected, because if BAX and BAK pores are not selective and if they allow the release of molecules like mtDNA, they should also release mtRNA, which limits the novelty of the findings. On the other hand, it is important to show that mtRNA is indeed also released and that in this case it contributes to the inflammatory responses of the SASP via the cytosolic RNA sensors. Yet, the use of BAX/BAK depletion to study the role of mtRNA in SASP in vivo is not optimal, since it also blocks mtDNA release and the effects are not disentangled in this model. In this sense, a key question that the authors should address is the relative contribution of mtRNA/MAVS and mtDNA/STING to the SASP.

Other major concerns are related to the need of additional experiments to substantiate the findings:

- In the microscopy experiments detecting mtRNA in the cytosol, a positive control showing mtRNA in mitochondria in proliferative cells is missing, so that it can be concluded that it is indeed not released.
- In the experiments with transfection of mtRNA in cells without mitochondria, the induction of SASP should be compared with transfection with mtRNA as well as with mtRNA+mtDNA.
- In the experiments with POLRMT the authors should confirm that the treatment has not affected mtDNA levels and that mtDNA is still released to the cytosol, which would also affect the SASP.
- The experiments in Figure 3 with RIG-I and MDA5 depletion should also be compared with TLR3 depletion.
- Since BAX and BAK depletion also blocks mtDNA release, the authors should emphasize that the results obtained regarding the SASP are also related to this effect. At the moment, the text is not sufficiently clear in this respect.
- While the use of in vivo models of BAX/BAK depletion in MASH extends the role of these proteins in SASP in the liver, this model is not optimal to support the role of mtRNA in this process. Here, depleting MAVS would be more relevant to support the role of mtRNA on SASP. For the MASH model, the authors should analyze the release of mtDNA and the activation of the STING pathway. Using STING inhibitors would also help to disentangle the contribution of mtDNA and mtRNA to the SASP in these settings.

Minor points:

- WBs should be quantified.
- The increased levels of p21 and p16 in Figure 1 should also be shown at the protein level.
- In Figure 2, the increased levels of MDA5, RIG-I and TLR3 should be shown at the protein level, with quantified WBs.
- The activation of the MAVS pathway downstream of RIG-I and MDA5 is expected and the results in Figure 4 should be merged with Figure 3 and/or moved to supplementary.

Version 1:

Reviewer comments:

Reviewer #1

(Remarks to the Author)

Thanks for your clarifications, my concerns have been met.

Reviewer #2

(Remarks to the Author)

The authors addressed most of the reviewers' concerns adequately and provided convincing explanations for the questions that could not be solved.

We are very grateful for the constructive and thoughtful feedback from the reviewers, whose suggestions have greatly strengthened our work. We have addressed each question to the best of our ability, and as a result, we believe the revised manuscript is both more robust and substantially improved. We are proud of the quality and rigor of the data now presented.

Reviewer #1 (Remarks to the Author):

The work phenocopies studies by this group showing the role of mtDNA release for cGAS and STING activation as a driver of the SASP, and is well in line with a recently released paper by the Serrano group, also published in Nature Communications, early last week?! PMID: 39191740

The authors convincingly show that mitochondria-derived dsRNA are found at increased levels in the cytoplasm of senescent cells. This can activate cytosolic RNA sensors (known) to drive the senescence associated secretory phenotype, but is unrelated to the senescence induced cell cycle block via p53, p16, p21. They go on and show that loss of cytosolic RNA sensors RIGI and MDA5 or MAVS (but not TLR3) are important for the inflammatory Nf- κ B signature found in senescent cells. Release of mtdsRNA requires the BCL2 family proteins BAX and BAK (based on using double-deficient cells as reference; a finding reported before). Finally, they show in a model of western-diet induced liver ageing, dsRNA sensors are increased and their expression as well as the secretory senescence phenotypes are reduced on a BAX/BAK mutant background.

The study is convincing, the experiments well done. I have only few concerns.

We thank the reviewer for their positive and thoughtful assessment of our work.

1) the authors show that both mtDNA and mtdsRNA can drive the SASP and the data looks very clean in both studies. However, one would expect some redundancy. Does combined loss of MAVS + cGAS (or STING) act in an additive manner - as one would expect, or are they fully redundant? This can be tested in one of the model systems by assessing the impact of combined vs. single deletion of MAVS and or cGAS (or STING).

That is an excellent question. To address this directly, we performed the following experiments: First, we compared the effects of MAVS knockdown with STING inhibition in senescent cells. In our hands, pharmacological inhibition of STING using SN-011 led to a robust suppression of multiple SASP cytokines, whereas MAVS knockdown resulted in a more modest reduction. Notably, combined treatment did not produce additive suppression (Extended Data Figure 8). These findings suggest that MAVS and STING converge on shared downstream pathways or act redundantly, with cGAS-STING playing the dominant role in SASP regulation. Second, we used the Parkin-mediated mitophagy system to remove mitochondria from senescent cells and then transfected them with either mtDNA or mtRNA (2.5 μ g) purified from isolated mitochondria. We found that both mtDNA and mtRNA could partially restore SASP expression globally; however, some SASP factors were shared between both nucleic acids, while others were unique to each (see new Extended Data Figure 4). Together, these findings suggest that the MAVS and cGAS-STING pathways are not fully redundant but regulate overlapping yet distinct components of the SASP.

2) Work by the Garcia-Saez group suggests that BAX and BAK influence each other, coordinating the growth of the "apoptotic pore" with Bax forming larger structures compared to BAK - is there any reason to believe that mtDNA vs. mtRNA may be released preferentially through one or the other, or are they fully redundant (discuss / maybe test Bax vs BAK single KO cells in one of the paradigms (e.g. cytokine mRNA induction))

Thank you for this insightful comment. As suggested by the reviewer, we have performed experiments using siRNA to knock down BAX and BAK individually and in combination (see new Extended Data Figure 9). Interestingly, knockdown of BAX alone resulted in a significant reduction in the expression of common SASP factors, including IL-6, IL-8, and IL-1 β . In contrast, BAK knockdown had no effect. Combined knockdown of both BAX and BAK showed a trend toward further reduction, but the strongest individual effect was observed with BAX inhibition.

These findings suggest that BAX plays a more prominent role than BAK in regulating the SASP, at least in our system. While this implies that BAX may be more directly involved in the release of mitochondrial components that trigger inflammatory signaling, we have not yet determined whether the observed effects are due to preferential release of mtDNA, mtRNA, or both. Further experiments will be required to clarify whether BAX and BAK differentially regulate the release of specific mitochondrial nucleic acids during senescence.

3) The MASH data is correlative at best, as it does not show that diet-induced SASP phenotypes are driven by any of the RNA sensors or MAVS. This is my biggest worry, as it actually only shows dsRNA sensors are up in MASH mice, and BAX/BAK DKO mice show reduced damage/SASP. It could equally well be cGAS/STING mediated (expression can easily be elevated, too), and both may contribute to the phenotype (or lack thereof). So, I suggest to assess levels of cGAS/STING and signalling components downstream (TBK1 and IRF3) in parallel and adapt/discuss findings depending on outcome. I do not ask the authors to repeat these experiments using siRNAs against any of these components at that stage (even though it would be the relevant controls)

We thank the reviewer for this important comment. We agree that the MASH data in the original submission are correlative and do not yet provide definitive mechanistic proof that the observed SASP phenotypes are driven exclusively by RNA sensing via MAVS. However, we have taken several steps to strengthen our interpretation and address this issue experimentally.

Using Proximity Ligation Assay (PLA), we found increased co-localization of RIG-I and MAVS in the livers of FFC-fed mice, consistent with activation of MAVS signaling during MASH. Importantly, liver-specific deletion of BAX and BAK markedly reduced RIG-I/MAVS colocalization (new Extended Data Fig. 10f), indicating that miMOMP is upstream of MAVS activation in this setting.

RNA-sequencing from livers of FFC-fed BAX/BAK double knockout mice revealed a significant downregulation of both RIG-I-like receptor signaling and cytosolic DNA sensing pathways. These data suggest that BAX/BAK-dependent mitochondrial stress activates both RNA- and DNA-mediated innate immune responses in MASH. Following the reviewer's suggestion, we attempted to measure phospho-TBK1 and phospho-IRF3 levels in liver tissue by western blot. Despite

repeated attempts, we were unable to obtain reliable or reproducible signals, likely due to technical limitations in detecting these phospho-proteins in vivo.

To directly test the role of MAVS, we performed liver-specific knockdown of MAVS in FFC-fed mice. This intervention significantly reduced SASP factor expression, decreased immune cell infiltration, and improved liver function, as reflected by lower serum AST and ALT levels (new Extended Data Fig. 11).

Taken together, these results support a functional role for MAVS-dependent signaling in promoting inflammation and tissue damage in MASH, while not excluding a contribution from cGAS–STING. We now highlight these limitations in the revised manuscript and explicitly note that future studies will be required to dissect the relative contributions of mtDNA- versus mtRNA-mediated signaling in MASH.

4) the authors may also discuss a recent study showing that mtDNA expression levels correlate with the proliferative state in fibroblasts and are reported to be very low in quiescent cells, while high upon their immortalization/transformation (PMID: 39209534). Another study misses to detect colocalization of MDA5 with mtDNA in the cytosol (PMID: 38955468) These observations don't add up with what is shown here and should be discussed, along with the role of VDACs in this process, in light of their own observations.

Thank you for highlighting these studies. PMID: 39209534 found low mt-dsRNA levels in quiescent fibroblasts and higher levels in immortalized or transformed cells. In contrast, we find that senescent cells, although non-proliferative, accumulate mt-dsRNA and activate downstream RNA sensing pathways. This difference likely reflects fundamental distinctions between quiescence and senescence, particularly in mitochondrial function and the activation of BAX and BAK.

Regarding the second study (PMID: 38955468), which found that mtDNA primarily localizes with stress granules and shows little co-localization with MDA5, we acknowledge this discrepancy and now discuss it in the manuscript. The authors of that work proposed that MDA5 binding to mt-dsRNA may be transient. In our case, RNA immunoprecipitation demonstrates that mtDNA binds stably to MDA5 in senescent cells, and MDA5 deletion reduces SASP expression, supporting a functional interaction in this context. We argue that this difference may reflect variations in cell type, how mtDNA accumulates, or the timing at which MDA5–RNA interactions are captured.

Finally, we have expanded our discussion on the potential role of VDACs in mediating the cytosolic release of mtDNA, integrating this with our observations. While VDACs have been implicated in the cytosolic release of mtDNA in senescence and aging (PMID: 37532932, 40203808), to our knowledge their role in release of mtDNA has not been demonstrated.

Reviewer #2 (Remarks to the Author):

The authors report a role of mtRNA on the SASP during senescence. They identify the release of mtRNA into the cytosol in senescent cells, which they associate with an increase in MDA5 and RIG-I expression and the activation of MAVS. They also show that cytosolic mtRNA can induce a SASP response, independent of other mitochondrial components. In line with the previous reported role of BAX and BAK on the release of mtDNA, they report that mtRNA release in senescence also depends on these proteins. And they show that BAX and BAK contribute to the SASP in the liver during MASH.

On the one hand, the results are kind of expected, because if BAX and BAK pores are not selective and if they allow the release of molecules like mtDNA, they should also release mtRNA, which limits the novelty of the findings. On the other hand, it is important to show that mtRNA is indeed also released and that in this case it contributes to the inflammatory responses of the SASP via the cytosolic RNA sensors. Yet, the use of BAX/BAK depletion to study the role of mtRNA in SASP *in vivo* is not optimal, since it also blocks mtDNA release and the effects are not disentangled in this model. In this sense, a key question that the authors should address is the relative contribution of mtRNA/MAVS and mtDNA/STING to the SASP.

We agree with the reviewer that using BAX/BAK knockout mice *in vivo* does not allow us to fully separate the respective contributions of mtDNA and mtRNA signaling to inflammation during MASH, as these proteins are likely permissive for the release of both nucleic acid species. To begin addressing this, we performed several complementary experiments. First, using proximity ligation assay (PLA), we demonstrated increased co-localization of RIG-I and MAVS in liver tissue from FFC-fed mice, consistent with activation of RNA sensing pathways during MASH. In BAX/BAK-deficient livers, this RIG-I/MAVS co-localization was markedly reduced, indicating diminished MAVS activation. Second, transcriptomic analysis of livers from FFC-fed BAX/BAK-deficient mice revealed significant downregulation of both RNA sensing and DNA sensing pathways, suggesting that both arms of mitochondrial nucleic acid signaling are engaged under MASH conditions and may each contribute to the phenotype. Third, to directly test the functional role of mtRNA sensing *in vivo*, we performed liver-specific knockdown of MAVS in FFC-fed mice. This intervention significantly reduced SASP factor expression, decreased immune cell infiltration, and improved liver function, as measured by reduced serum AST and ALT levels. These findings provide clear evidence that RNA sensing via MAVS contributes to inflammation and tissue damage in MASH. While we acknowledge that a direct head-to-head comparison of mtRNA/MAVS and mtDNA/cGAS-STING signaling *in vivo* has not yet been performed, our combined *in vivo*, transcriptomic, and imaging data suggests that both pathways are activated in MASH and that MAVS-dependent RNA sensing plays a functionally important role.

Other major concerns are related to the need of additional experiments to substantiate the findings:

-In the microscopy experiments detecting mtRNA in the cytosol, a positive control showing mtRNA in mitochondria in proliferative cells is missing, so that it can be concluded that it is indeed not released.

In proliferative cells, we see minimal (or low levels of) J2 staining inside mitochondria or cytosol. The reason is that mitochondria transcribe both the heavy and light strands of mtDNA almost entirely, but transcription is generally asynchronous, reducing the chance of extensive complementary overlap that would generate stable dsRNA. The low J2 signal inside mitochondria in both proliferative and senescent cells is consistent with the expectation that any detectable mtdsRNA likely accumulates in the cytosol, either due to leakage or impaired degradation.

-In the experiments with transfection of mtRNA in cells without mitochondria, the induction of SASP should be compared with transfection with mtRNA as well as with mtRNA+mtDNA.

We thank the reviewer for this excellent suggestion. To address this point, we performed additional experiments in which we transfected senescent cells depleted of mitochondria using the Parkin-mediated mitophagy system with either mtRNA, mtDNA, or both. We first optimized conditions by testing different concentrations of each nucleic acid, which led us to use 2.5 μg for subsequent experiments.

RNA sequencing of these cells (new Extended Data Figure 4) showed that transfection with either mtRNA or mtDNA alone partially restored the SASP phenotype. Across a broad panel of inflammation-related genes, including multiple IFNB1 and NF- κB targets, as well as SenMayo, some SASP genes were induced by both nucleic acids, likely reflecting convergence of MAVS and cGAS–STING signaling on shared downstream transcription factors, while others responded exclusively to mtRNA or to mtDNA. Unexpectedly, co-transfection of both mtRNA and mtDNA did not produce an additive effect; in most cases, expression levels were lower when both were combined (see q-PCR data for IL-8), a trend that

was consistent across several SASP markers. One possible explanation is that both pathways feed into common downstream signaling components, creating a bottleneck when both are activated simultaneously and limiting maximal NF- κB activation. Given the complexity and ambiguity of this result, we have chosen not to include it in the manuscript but acknowledge it here in response to the reviewer’s question. We decided to focus on the comparison between the inflammatory responses to mtRNA and mtDNA, which we believe is informative and reveals important differences in how mitochondrial RNA and DNA contribute to SASP regulation, while also highlighting that these pathways, although distinct, ultimately converge on shared inflammatory signaling mechanisms.

-In the experiments with POLRMT the authors should confirm that the treatment has not affected mtDNA levels and that mtDNA is still released to the cytosol, which would also affect the SASP.

We thank the reviewer for this important point. To address this, we performed immunostaining for TOM20 and DNA and quantified extramitochondrial DNA signals. Our analysis showed that

treatment with the POLRMT inhibitor IMT1 did not alter cytosolic mtDNA levels (new Extended Data Fig. 5). This is consistent with recent findings from the Serrano lab, indicating that that POLRMT inhibition specifically reduces mitochondrial RNA synthesis without impacting mtDNA release (PMID: 39191740).

-The experiments in Figure 3 with RIG-I and MDA5 depletion should also be compared with TLR3 depletion.

We thank the reviewer for this suggestion. To further address this point, we now performed siRNA-mediated knockdown of TLR3. Consistent with the inhibitor data, TLR3 depletion did not significantly alter SASP gene expression (new Extended Data Figure 6e-g). Together, these results indicate that, in the context of senescence, RNA sensing is primarily mediated by MDA5 and RIG-I rather than endosomal TLR3.

-Since BAX and BAK depletion also blocks mtDNA release, the authors should emphasize that the results obtained regarding the SASP are also related to this effect. At the moment, the text is not sufficiently clear in this respect.

Thank you for this helpful comment. We agree that this point should be stated more clearly. We have revised the text to explicitly acknowledge that the effects of BAX/BAK depletion on the SASP likely reflect reduced activation of both RNA-sensing (MAVS) and DNA-sensing (cGAS-STING) pathways. We have also performed additional experiments aimed at beginning to dissect their relative contributions which are now included in the revised version.

-While the use of in vivo models of BAX/BAK depletion in MASH extends the role of these proteins in SASP in the liver, this model is not optimal to support the role of mtRNA in this process. Here, depleting MAVS would be more relevant to support the role of mtRNA on SASP. For the MASH model, the authors should analyze the release of mtDNA and the activation of the STING pathway. Using STING inhibitors would also help to disentangle the contribution of mtDNA and mtRNA to the SASP in these settings.

Thank you for this thoughtful comment. We agree that the BAX/BAK knockout model, while useful in demonstrating the broader role of MOMP in driving SASP and inflammation during MASH, does not allow us to isolate the specific contribution of mtRNA signaling. As the reviewer rightly points out, BAX/BAK depletion prevents the release of both mtDNA and mtRNA, making it difficult to disentangle the roles of each nucleic acid species. To address this limitation, we have complemented the liver specific BAX/BAK KO model with a liver-specific knockdown of MAVS in the MASH setting (see new Extended Data Figure 11). This intervention led to a significant reduction in SASP factor expression, immune cell infiltration, and liver function, supporting a role for mtRNA and MAVS-dependent signaling in this context. We agree that further analysis of mtDNA release and cGAS-STING pathway activation in the MASH model would strengthen these findings. We have revised the manuscript to clearly acknowledge these limitations and to highlight the need for additional experiments to dissect the relative contributions of mtDNA and mtRNA signaling in vivo.

Minor points:

-WBs should be quantified.

We have quantified all western blots.

-The increased levels of p21 and p16 in Figure 1 should also be shown at the protein level.

We have done so (Extended Data Figure 1).

-The activation of the MAVS pathway downstream of RIG-I and MAD5 is expected and the results in Figure 4 should be merged with Figure 3 and/or moved to supplementary.

We considered the reviewer's suggestion and attempted to merge the MAVS data with Figure 3. However, combining them made the figure overly complex and difficult to follow. While RIG-I and MDA5 both signal through MAVS, we felt it was important to highlight the MAVS knockdown data as a main figure because it directly demonstrates that MAVS is essential for SASP regulation in senescent cells, thereby functionally linking the upstream sensors to the downstream response.